# A Cryptochrome 2 mutation yields advanced sleep phase in humans

Arisa Hirano[1], Guangsen Shi[1], Christopher R Jones[2], Anna Lipzen[3,4], Len A Pennacchio[3,4], Ying Xu[5], William C Hallows[1], Thomas McMahon[1], Maya Yamazaki[1], Louis J Ptáček[1,6]*, Ying-Hui Fu[1]*

[1]Department of Neurology, University of California, San Francisco, San Francisco, United States; [2]Department of Neurology, University of Utah, Salt Lake City, United States; [3]Lawrence Berkeley National Laboratory, Berkeley, United States; [4]Department of Energy Joint Genome Institute, Walnut Creek, United States; [5]Center for System Biology, Soochow University, Suzhou, China; [6]Howard Hughes Medical Institute, University of California, San Francisco, San Francisco, United States

**Abstract** Familial Advanced Sleep Phase (FASP) is a heritable human sleep phenotype characterized by very early sleep and wake times. We identified a missense mutation in the human Cryptochrome 2 (*CRY2*) gene that co-segregates with FASP in one family. The mutation leads to replacement of an alanine residue at position 260 with a threonine (A260T). In mice, the *CRY2* mutation causes a shortened circadian period and reduced phase-shift to early-night light pulse associated with phase-advanced behavioral rhythms in the light-dark cycle. The A260T mutation is located in the phosphate loop of the flavin adenine dinucleotide (FAD) binding domain of CRY2. The mutation alters the conformation of CRY2, increasing its accessibility and affinity for FBXL3 (an E3 ubiquitin ligase), thus promoting its degradation. These results demonstrate that CRY2 stability controlled by FBXL3 plays a key role in the regulation of human sleep wake behavior.

*For correspondence: ljp@ucsf.edu (LJP); ying-hui.fu@ucsf.edu (Y-HF)

## Introduction

Sleep is vital for all animals. Sleep-wake timing is regulated by the internal biological clock driving physiological rhythms with a period of approximately 24 hr (*Takahashi, 1995*). The circadian clock is composed of interlocked transcriptional and translational negative feedback loops (*Lowrey and Takahashi, 2004*; *Reppert and Weaver, 2001*). In mammals, a CLOCK-BMAL1 heterodimer binds to E-boxes and activates gene expression of the *Period (Per)* and *Cryptochrome (Cry)* genes. Translated PERs and CRYs proteins form a complex that enters the nucleus to inhibit their own transcription through direct interaction with CLOCK-BMAL1 heterodimers. PER and CRY proteins accumulating in the nucleus are then degraded over time. As protein levels fall (depending on rate of degradation), the transcription-translation feedback loop begins anew.

*CRY2* is a principal component in mammalian circadian clocks (*Shearman et al., 2000*; *van der Horst et al., 1999*; *Vitaterna et al., 1999*). While Drosophila and plant CRY proteins act as photoreceptors contributing to photoentrainment of the circadian clock and other biological processes by binding to flavin adenine dinucleotide (FAD) (*Partch and Sancar, 2005*), mammalian CRY2 has light-independent transcriptional repressor activity and strongly inhibits E-box-regulated gene expression (*Griffin et al., 1999*; *Kume et al., 1999*; *Shearman et al., 2000*). The protein stability of CRY2 is fine-tuned by post-translational modification including phosphorylation and ubiquitylation. In addition, various enzyme modifications play a role in CRY2 regulation (*Reischl and Kramer, 2011*;

**eLife digest** Sleep is an essential process in animals. In humans, the disturbance of normal sleep-wake cycles through shift-work or long-term sleep disorders increases the risk of developing conditions including mental illness, cancer and metabolic syndromes. Understanding how sleep-wake behavior is controlled within cells may help researchers to develop effective therapies to reduce the ill effects of disturbed sleep-wake cycles on health.

To understand how our sleep-wake cycles are regulated in cells, researchers have been looking for genetic mutations that affect human sleep schedules. For example, some people have a 'morning lark' schedule that makes them prone to go to sleep early and rise early the next day. Others are prone to be 'night owls', staying up later at night and waking up later in the morning. By studying the mutations that underlie these behaviors, researchers hope to understand precisely how these genes regulate sleep schedules.

Now, Hirano et al. have identified a particular mutation in a gene called Cryptochrome 2 (*CRY2*) that causes people to have shorter sleep-wake cycles so that they wake up very early in the morning and struggle to stay awake in the evening. For the experiments, mice were genetically engineered to carry the mutant human *CRY2* gene, which shortened the sleep-wake cycles of the mice and their responses to light so that they both woke up earlier and went to sleep earlier.

Further experiments examined what effect the mutation has on the protein that is produced by *CRY2.* The mutation changes the shape of the protein, which allows an enzyme called FBXL3 to bind to the mutant protein more easily and rapidly break it down. The length of sleep cycles may be determined by how long it takes FBXL3 to break down the protein produced by *CRY2*. The findings of Hirano et al. may help researchers to develop treatments for people with sleep problems.

---

*Stojkovic et al., 2014*). Among them, FBXL3 is an F-box type E3 ubiquitin ligase which promotes CRY1 and CRY2 ubiquitylation thus leading to proteasome-mediated degradation (*Busino et al., 2007*). Mutations in mouse *Fbxl3* or knockout of the *Fbxl3* gene dramatically lengthens the period of mouse behavioral rhythms in constant darkness (*Godinho et al., 2007*; *Hirano et al., 2013*; *Shi et al., 2013*; *Siepka et al., 2007*), indicating that the protein stability of CRY1 and CRY2 is a critical determinant of circadian period in mice. However, direct evidence supporting the significance of *CRY2* and the post-translational regulation of CRY2 protein in the human circadian clock regulating the sleep-wake cycle has been lacking.

Familial Advanced Sleep Phase (FASP) is a heritable sleep phenotype characterized by stable early sleep and wake times (*Jones et al., 1999*; *Reid and Burgess, 2005*; *Reid et al., 2001*). The FASP phenotype can segregate as a highly penetrant, autosomal dominant trait in human kindreds. Previously, we have identified mutations in clock genes, including *Period2, Period3, casein kinase Iδ*, and *Dec2* causing circadian and sleep homeostasis phenotypes in humans (*He et al., 2009*; *Toh et al., 2001*; *Xu et al., 2005*; *Zhang et al., 2016*). A mutation at the phosphorylation priming site of PER2 attenuates sequential phosphorylation and consequently destabilizes PER2 proteins. The mouse model expressing mutant PER2 exhibits a shortened circadian period accompanied with large phase-advance in sleep-wake rhythms (*Xu et al., 2007*). This sequential phosphorylation region of PER2 was later found to be modulated by another post-translational regulation, *O*-GlcNAcylation, demonstrating an interplay and competition between phosphorylation and *O*-GlcNAcylation of serine residues in this region (*Kaasik et al., 2013*). These studies highlighted the important role of post-translational regulation of clock proteins in vivo in humans and also revealed mechanistic insight into the regulation of PER2. Thus, human genetic studies have provided valuable and unique opportunities to elucidate novel molecular mechanisms of circadian/sleep regulation.

Here we report the identification of a novel variant in the human h*CRY2* gene that leads to FASP. Generation of a mouse model carrying the mutation revealed that the mutation causes a FASP-like phenotype in mice with altered circadian period and photic entrainment. We found that the mutation in the CRY2 FAD-binding-domain enhances its affinity for FBXL3, thus destabilizing CRY2 via increased ubiquitylation and targeting for degradation by the proteasome. We conclude that

regulation of CRY2 stability by a proper balance of FAD and FBXL3 is essential for the sleep-wake cycle in humans.

## Results

### Identification of a novel mutation in the h*CRY2* gene associated with FASP

Through candidate gene screening in FASP families, we identified a missense mutation in the human *CRY2* gene, which causes an amino acid conversion from Ala→Thr at position 260 (A260T) (*Figure 1A*). No other novel mutations were found in ~25 candidate circadian genes that were sequenced. The A260T mutation is associated with the circadian phenotype in this FASP family (*Figure 1A*) (*Jones et al., 1999*; *Toh et al., 2001*; *Xu et al., 2005*; *Zhang et al., 2016*). The fraternal twin sisters inherited the mutation from their mother and both reported a strong morning preference (Horne-Ostberg scores of 84 and 72) (*Figure 1—source data 1*). The proband also had a very early melatonin onset (4:41 P.M.), while the averaged melatonin onset of normative samples is 8:50 P.M. (*Burgess and Fogg, 2008*). Her melatonin onset time is 3.35 standard deviations earlier than expected and among the earliest 0.05% of normative samples (*Burgess and Fogg, 2008*). Ala260 is located in the FAD binding domain of CRY2 and it is highly conserved in CRY1 and CRY2 proteins of various species (*Figure 1B*).

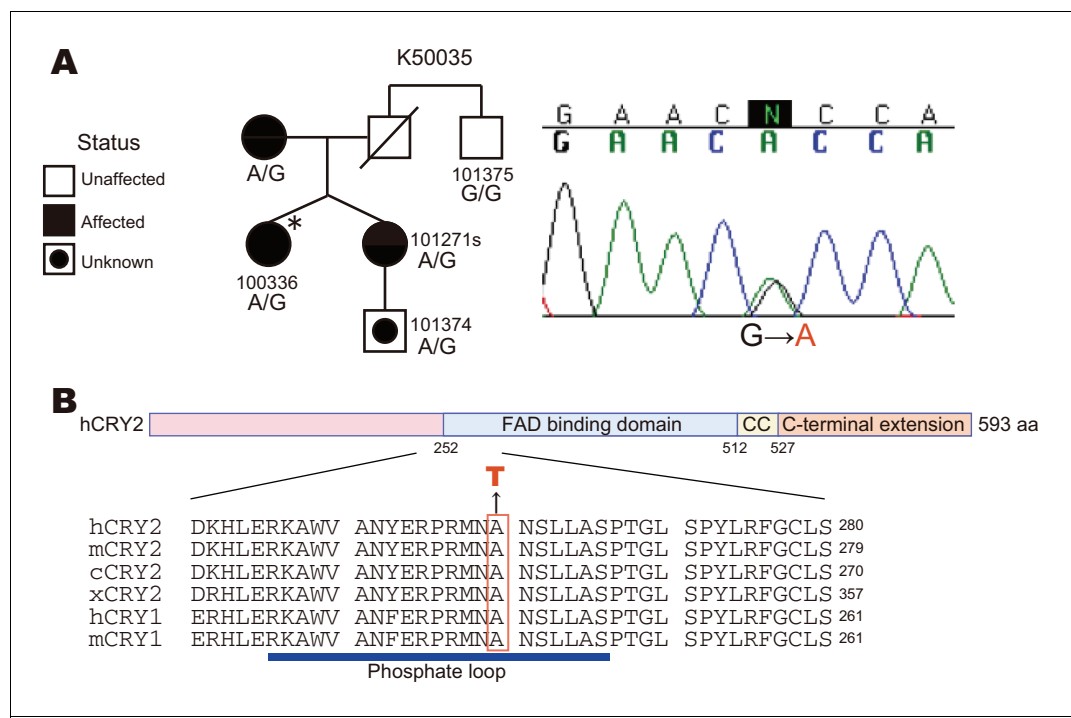

**Figure 1.** A *CRY2* mutation in FASP kindred 50035. (**A**) Pedigree of the family (kindred 50035) segregating the *CRY2* mutation (A260T). Circles and squares represent women and men, respectively. An asterisk marks the proband. A missense mutation from G to A causes an amino acid conversion from Alanine to Threonine at position 260. (**B**) Amino acid alignment around the mutation site. The A260T mutation is located in the N-terminal portion of the FAD binding domain in CRY2. This residue is highly conserved among vertebrate species. CC denotes a Coiled-Coil sequence.

The following source data is available for figure 1:

**Source data 1.** Summary of sleep phenotype of human subjects.

## FASP in mouse model carrying A260T mutation

To test whether the A260T mutation causes FASP and has a dominant effect on the circadian sleep-wake cycle, we generated wild-type hCRY2 (hCRY2-WT) and mutant hCRY2 (hCRY2-A260T) human BAC transgenic (Tg) mice (*Figure 2—figure supplement 1A*). Transgenic mice were subjected to locomotor behavioral analysis using a video camera tracking system. Under conditions of 12 hr light and 12 hr dark (LD 12:12), both hCRY2-WT and hCRY2-A260T mice entrained stably to the LD cycle (*Figure 2A*). However, the peak time of resting behavior, as determined by quadratic-function fitting, was significantly advanced in hCRY2-A260T mutant mice (*Figure 2B*). The activity offset and onset times were also advanced in hCRY2-A260T mutant mice vs. hCRY2-WT mice (*Figure 2C* and *Figure 2—figure supplement 1B*). Similarly, hCRY2-A260T mutant mice on a *Cry2* null background demonstrated advanced activity onset and offset, especially around the LD transition (ZT12-13) (*Figure 2—figure supplement 1C*). These results demonstrate that hCRY2-A260T mice recapitulate the advanced sleep phase seen in the human FASP subjects harboring the *CRY2* mutation.

## Shortened period and reduced phase-shift in hCRY2-A260T mice

We next analyzed voluntary wheel-running activity to evaluate phase-shift and free-running period of the circadian clock for the Tg mouse models. Similar to locomotor activity measured by video tracking (*Figure 2A*), wheel-running activity offset times were advanced in hCRY2-A260T vs. hCRY2-WT in LD on both mCry2 WT and null backgrounds (*Figure 2D*; *Figure 2—figure supplement 2*), while there are no significant differences in activity onset time and acrophase. Interestingly, hCRY2-A260T showed reduced phase-delay when mice were subjected to a 30-min light pulse at ZT14 (*Figure 2D, E*), whereas phase-advance was normal in response to a light pulse at ZT22 (*Figure 2—figure supplement 3*). Thus, mutant mice have reduced sensitivity to entrainment by light at early night compared to control mice. The mice were subsequently released into constant darkness (DD) to determine circadian period. The free-running period of hCRY2-A260T (23.52±0.04 hr) was significantly shorter than that of hCRY2-WT (23.70±0.03 hr) and WT mice (23.74±0.02 hr) (*Figure 2F*). The period shortening phenotype was further enhanced by crossing Tg mice onto the mCry2 null background (*Figure 2—figure supplement 4A*). The shorter circadian period and reduced phase-delay were also observed in another mutant line (23.51±0.06 hr) with a higher mutant transgene copy number (*Figure 2—figure supplements 1A*, *4B*), thus excluding the possibility that the phenotype was due to positional effects of the transgene insertion site in the genome. Of note, there is no significant difference in the periods and phase-shifting of hCRY2-WT transgenic vs. transgene negative mice (*Figure 2E,F*), indicating that the shortening of circadian period and abnormal phase-delay are not simply due to overexpression of hCRY2. Taken together, these data demonstrate that the phase-advances in mice and humans results from the *CRY2* mutation. Data from the transgenic mice suggests that the phase advance may be due to a combination of shortened period and altered sensitivity to photic entrainment.

## Shortened circadian period in peripheral clocks of hCRY2-A260T

The effect of the mutation on the circadian period and phase angle in the peripheral clock was examined by crossing BAC transgenic mice with mPer2^LUC knock-in mice (*Yoo et al., 2004*). Consistent with the behavioral rhythms, shortened clock period was observed in PER2::LUC bioluminescence rhythms of liver and lung cultures from hCRY2-A260T vs. WT mice (*Figure 3A,B*). The peak and trough time of the bioluminescence rhythms were advanced in both tissues of hCRY2-A260T mice, suggesting that phase of the peripheral clock is also advanced by the mutation in vivo (*Figure 3C*). Circadian period shortening by the A260T mutation was also found using mouse embryonic fibroblasts (MEFs) derived from mice with a mutant vs. WT transgene on both WT and mCry2 knockout backgrounds (*Figure 3D,E*). In addition, NIH3T3 cells stably expressing CRY2-A260T also displayed a shorter circadian period than CRY2-WT expressing cells (*Figure 3F*), emphasizing the dominant effect of CRY2-A260T on the circadian period. Our results indicate that CRY2-A260T shortens the circadian period in both central and peripheral clocks, consistent with current understanding that core clock genes such as *Cry2* influence physiologies in multiple mammalian organ systems.

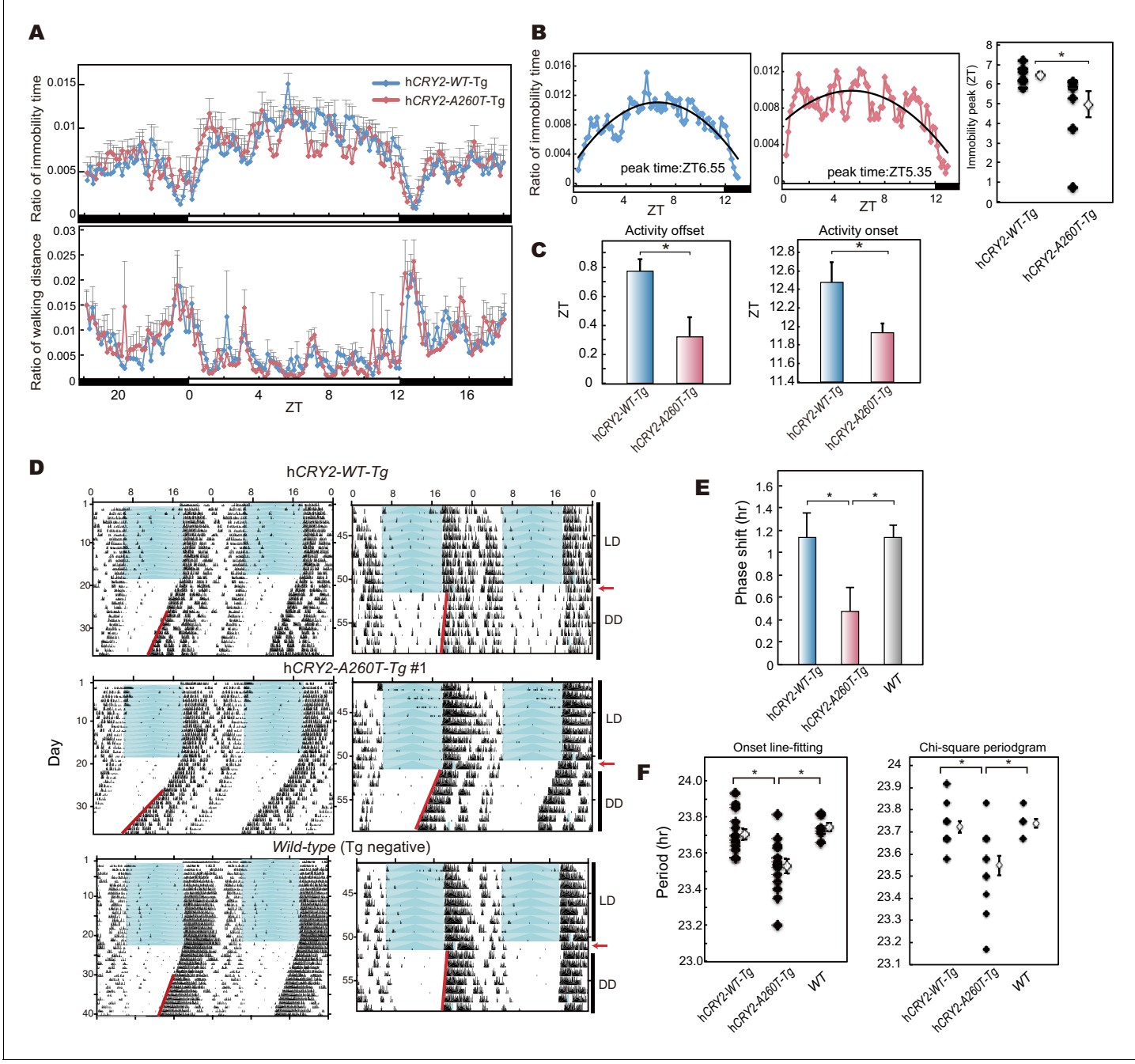

**Figure 2.** h*CRY2-A260T* mice have advanced phase of sleep-wake behavior in a light-dark cycle and a shortened circadian period in constant darkness. (A) Mouse movement was tracked by an infrared video camera in LD. The ratio of immobilization time to total daily immobilization time (upper panel) and the ratio of walking distance to total daily distance (bottom panel) were plotted every 10 min. Data are shown as means with SEM (n = 8 for h*CRY2-WT* and h*CRY2-A260T*). (B) Peak time of immobility was measured by fitting a quadratic function to data from ZT0 to 13. Representative examples of curve fitting for h*CRY2-WT* and h*CRY2-A260T* are shown here. Data are shown as means with SEM (n = 8 for h*CRY2-WT* and h*CRY2-A260T*, *p<0.05 by Student's *t*-test). (C) Onset and offset of locomotor activity. Data are shown as means with SEM (n = 8 for h*CRY2-WT* and h*CRY2-A260T*). (D) Actograms of wheel-running activity for *hCRY2-WT*, h*CRY2-A260T*, and littermate transgene-negative mice. The blue shadows indicate periods when the lights were on. Red lines were fitted to activity onset using ClockLab analysis software. (E) Phase-shifts in response to a 30-min light exposure at ZT14 indicated by red arrows in (D). *p<0.05 by Tukey's test (n = 7 for h*CRY2-WT*, n = 11 for h*CRY2-A260T*, n = 10 for WT). (F) The distribution of period measurements for BAC transgenic mice and transgene negative controls. Period was determined by line fitting of activity onset and chi-square periodogram from day 7 to day 19 in DD. *p<0.05 (n = 15 for h*CRY2-WT*, n = 14 for h*CRY2-A260T* and n = 7 for WT)

The following figure supplements are available for figure 2:

*Figure 2 continued on next page*

**Figure supplement 1.** Locomoter activity of h *CRY2* BAC Tg mice.
DOI: 10.7554/eLife.16695.006

**Figure supplement 2.** Wheel-running activity of h*CRY2* BAC Tg mice.
DOI: 10.7554/eLife.16695.007

**Figure supplement 3.** Light phase-shift of h*CRY2* BAC Tg mice at ZT22.
DOI: 10.7554/eLife.16695.008

**Figure supplement 4.** Wheel-running activity of h*CRY2Cry2* BAC Tg mice on a m KO background and another BAC Tg mice line.
DOI: 10.7554/eLife.16695.009

## Destabilization of CRY2 protein by the A260T mutation

The CRY2 Ala260 residue resides in the 'phosphate loop' responsible for binding to the phosphate of FAD (*Hitomi et al., 2009*) (*Figure 1B*). Mutations at amino acid residues critical for FAD binding affect CRY2 repressor activity of E-box-mediated transcriptional activation (*Czarna et al., 2013*; *Hitomi et al., 2009*; *Sanada et al., 2004*). Furthermore, Ser265 of mouse CRY2 (homologous residue of Ser266 in human CRY2) is a phosphorylation site, and the S265D mutation, mimicking a phosphorylated serine 265, reduces CRY2 repressor activity (*Sanada et al., 2004*). Using a Luciferase assay, we found that the A260T mutation weakened CRY2 repressor activity on *Per1* E-box-mediated transcriptional activation by CLOCK-BMAL1 (*Figure 4A*). However, the nuclear CRY2 protein levels in culture cells were decreased by the mutation (*Figure 4B*), which could potentially account for the reduction of CRY2 repressor activity (*Figure 4A*). We then examined the degradation of CRY2 proteins by cycloheximide (CHX) chase experiments. Consistent with the cellular distribution of CRY2 (*Figure 4B*), CRY2-A260T was less stable than CRY2-WT in HEK293 cells, especially in the nucleus (*Figure 4C*). Destabilization of CRY2 by the mutation was further verified in a CRY2-LUC based bioluminescence degradation assay, where the protein decay rate can be determined by recording CRY2-LUC bioluminescence in culture (*Hirano et al., 2013*; *Hirota et al., 2012*) (*Figure 4— figure supplement 1A*). Although Ala260 does not directly bind to FAD (*Hitomi et al., 2009*), amino acid conversion from the hydrophobic and small amino acid, alanine, to threonine could alter the conformation of the phosphate loop. This idea is supported by an observation that a mutation from alanine to aspartic acid (A260D) caused a more severe effect on CRY2 repressor activity and protein stability than the A260T mutation (*Figure 4—figure supplement 1A,B*). We found that human CRY1 harboring the corresponding mutation at position 241 is less stable than CRY1-WT, suggesting a common regulatory mechanism for CRY1 and CRY2 by FAD binding (*Figure 4—figure supplement 1C*). These results indicate that the conformation of the phosphate loop may play critical roles in regulating CRY2 stability and repressor activity.

## A260T mutation affects FBXL3-CRY2 interaction

FBXL3 primarily localizes in the nucleus and promotes proteasomal degradation of CRY2, consequently having a strong impact on the circadian period of mice (*Busino et al., 2007*; *Godinho et al., 2007*; *Siepka et al., 2007*; *Stojkovic et al., 2014*). A previous structural study demonstrated that C-terminal region of FBXL3 interacts with CRY2 through the FAD binding pocket and mutations in the FAD binding domain alter CRY2-FBXL3 interaction (*Xing et al., 2013*). We therefore speculated that the A260T mutation affects the FBXL3-CRY2 interaction, thus altering CRY2 protein stability. We first examined WT and mutant CRY2 stability in the absence of FBXL3. As anticipated, h*FBXL3* knockdown in HEK293 cells increased the stability of both CRY2-WT and CRY2-A260T. Interestingly, the destabilizing effect of the A260T mutation was abrogated by h*FBXL3* knockdown (*Figure 5A*), suggesting the effect of the mutation requires FBXL3.

   FAD stabilizes CRY2 by structurally interfering with the interaction between FBXL3 and CRY2 (*Xing et al., 2013*). We found that treatment of HEK293 cells with FAD increased CRY2-WT protein levels much more than CRY2-A260T (*Figure 5B*), suggesting that stabilization of CRY2 by FAD was reduced by the mutation. This result supports the hypothesis that the A260T mutation alters the FBXL3-CRY2 interaction. We thus carried out a competitive assay by adding FAD to complexed CRY2-FBXL3 to examine the effect of the mutation on the release of CRY2 from purified CRY2-FBXL3 complexes in vitro. Free CRY2-WT protein levels increased with addition of FAD in a dose-

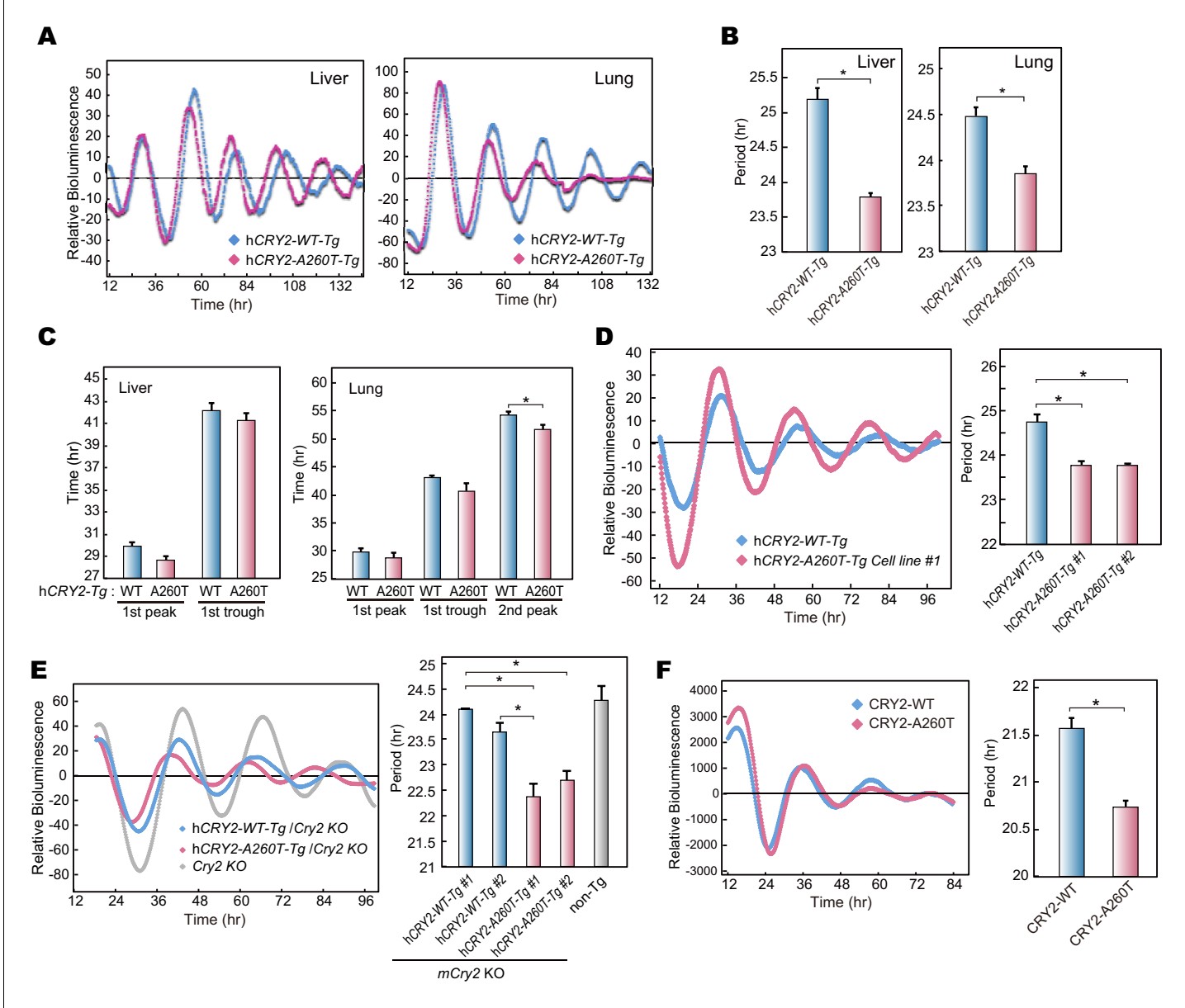

**Figure 3.** Bioluminescence rhythms in tissue cultures from h*CRY2-A260T* mice and *CRY2-A260T* stable cell lines. (**A**) Representative rhythms of PER2::LUC bioluminescence in the liver and lung. Data were detrended by subtracting the 24 hr average of bioluminescence. (**B**) Period measurements of the bioluminescence rhythms in liver and lung tissues. Data are shown as means ± SEM (n = 4 for liver and h*CRY2-WT* lung, n = 3 for h*CRY2-A260T* lung, *p<0.05 by Student's *t*-test). (**C**) Peak and trough time of PER2::LUC bioluminescence rhythms of mouse liver and lung tissues. Data are shown as means ± SEM (n = 4 for liver). Data are shown as means ± SD (n = 2 to 4 for lung, *p<0.05 by Student's *t*-test). (**D**) Representative rhythms of PER2::LUC bioluminescence in MEFs from h*CRY2* transgenic mice. Data were detrended by subtracting the 24 hr average of bioluminescence. Period lengths of the bioluminescence rhythms are shown as means ± SEM (n = 4, *p<0.05 by Tukey's test). (**E**) Representative examples of bioluminescence rhythms of m*Bmal1-luc* in MEFs from transgenic mice on a m*Cry2* knockout background. Cells were transfected with m*Bmal1*-luc vector 24 hr before the recording. Data were detrended by subtracting the 24 hr average of bioluminescence. Period lengths of the bioluminescence rhythms in the stable cell lines are shown as means ± SEM (n = 4, *p<0.05 by Games-Howell test). (**F**) Representative examples of bioluminescence rhythms of *Bmal1-luc* in NIH3T3 cells stably expressing FLAG-CRY2-WT or FLAG-CRY2-A260T. Data were detrended by subtracting the 24 hr average of bioluminescence. Period lengths of the bioluminescence rhythms in the stable cell lines are shown as means ± SEM (n = 3, *p<0.05 by Student's *t*-test).

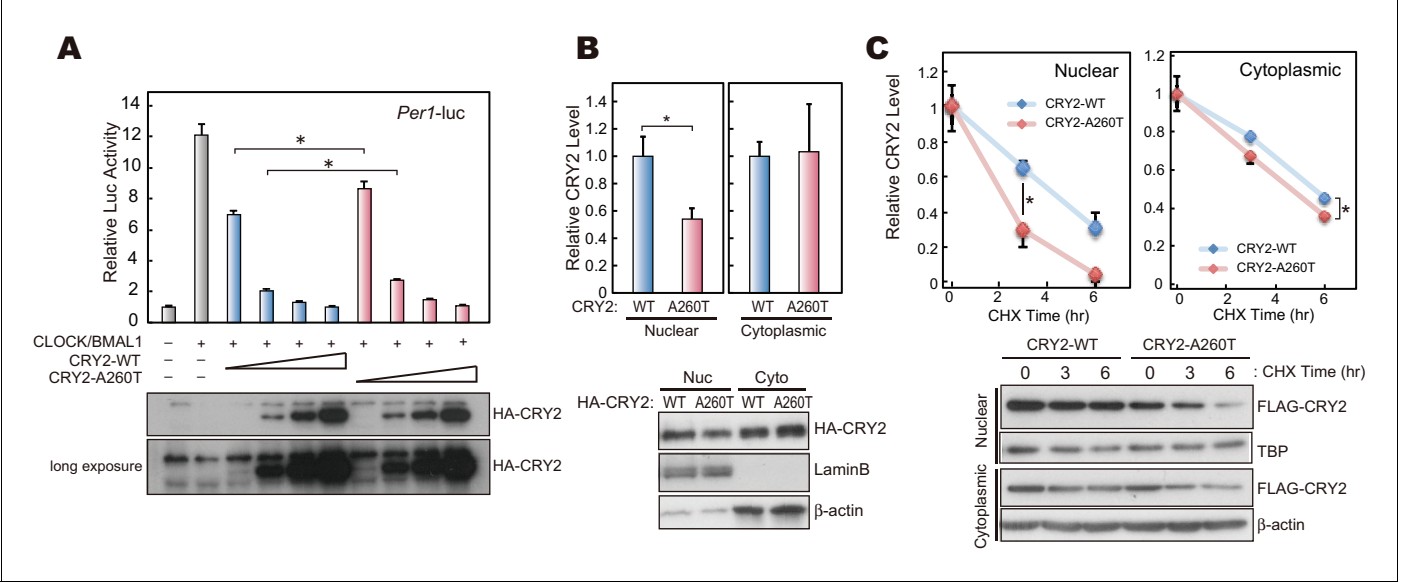

**Figure 4.** CRY2-A260T is less stable, particularly in nuclei. (**A**) Luciferase activity driven by mouse *Per1* E-box in HEK293 cells. 2, 5, 10 and 20 ng of hCRY2 expression vector (WT or A260T) was transfected into cells cultured in a 24-well plate. Luciferase activity was normalized to Renilla luciferase activity. Data are shown as means with SEM (n = 4, *p<0.05 by Student's *t*-test). (**B**) Protein levels of CRY2-WT and CRY2-A260T in the nuclear and cytosolic fractions of HEK293 cells. Data are shown as means with SEM (n = 3, *p<0.05 by Student's *t*-test). (**C**) Degradation assay of CRY2 protein in HEK293 cells. Forty-eight hours after transfection, cells were treated with 100 μg/ml CHX and fractionated into the nuclear and cytosolic fractions. CRY2 protein levels at the starting point (t = 0 hr) were normalized to 1. Data are shown as means with SEM (n = 3*, p<0.05 by Student's *t*-test).

The following figure supplement is available for figure 4:

**Figure supplement 1.** The effects of mutations at A260 on protein stability, repressor activity.

dependent manner (*Figure 5C*). In contrast, mutant CRY2 was released less readily than CRY2-WT from the complexes even though both forms of CRY2 were bound to FBXL3 at the same level (*Figure 5C*). In addition, we tested FBXL3-mediated CRY2 degradation using KL001, a small synthetic molecule known to stabilize CRY1 and CRY2 (*Hirota et al., 2012*) due to its structural similarity to FAD (*Nangle et al., 2013*). KL001 stabilizes CRY2-WT in a dose-dependent manner as was previously reported (*Hirota et al., 2012*). However, KL001 failed to stabilize CRY2-A260T to the same extent it did for CRY2-WT (*Figure 5D*, *Figure 5—figure supplement 1A*). These results indicate that the A260T mutation weakened the function of FAD and KL001 as inhibitors of FBXL3-mediated degradation of CRY2 and that CRY2-A260T is less stable than CRY-WT, likely due to strengthened interaction with FBXL3. To determine whether the A260T mutation indeed modifies the interaction of CRY2 and FBXL3, we performed co-immunoprecipitation analysis. As expected, CRY2-A260T binds more strongly to FBXL3 than CRY2-WT in HEK293 cells (*Figure 5E*), consequently leading to more ubiquitylation of the mutant protein (*Figure 5F*). The effect of the mutation on the binding affinity under in vivo conditions will need to be further evaluated when better human CRY2 antibodies (for immunoprecipitation) become available.

We next performed structural modeling of mutant CRY2 to address how the A260T mutation modulates conformation of the phosphate loop. For modeling, we used the mouse CRY2 structure as the crystal structure of mouse CRY2-FBXL3 complex is available (*Xing et al., 2013*) and amino acid sequence in the phosphate loop perfectly conserved (*Figure 1B*). The published CRY2-FBXL3 structure (*Xing et al., 2013*) revealed that space between Ala260 (corresponding to A259T in mouse) and Asp442 (mouse Asp441) in FBXL3-binding form (red) is more opened vs. the FAD-binding form (orange, *Figure 5—figure supplement 1B*). The amino acid change from Ala260 to Thr increases molecular density in this space and may alter electrostatic interactions between Ala260Thr and Asp442. As a result, the mutation likely renders the CRY2-A260T more accessible to FBXL3 binding (*Figure 5—figure supplement 1C*). This model is consistent with the results from CRY2-

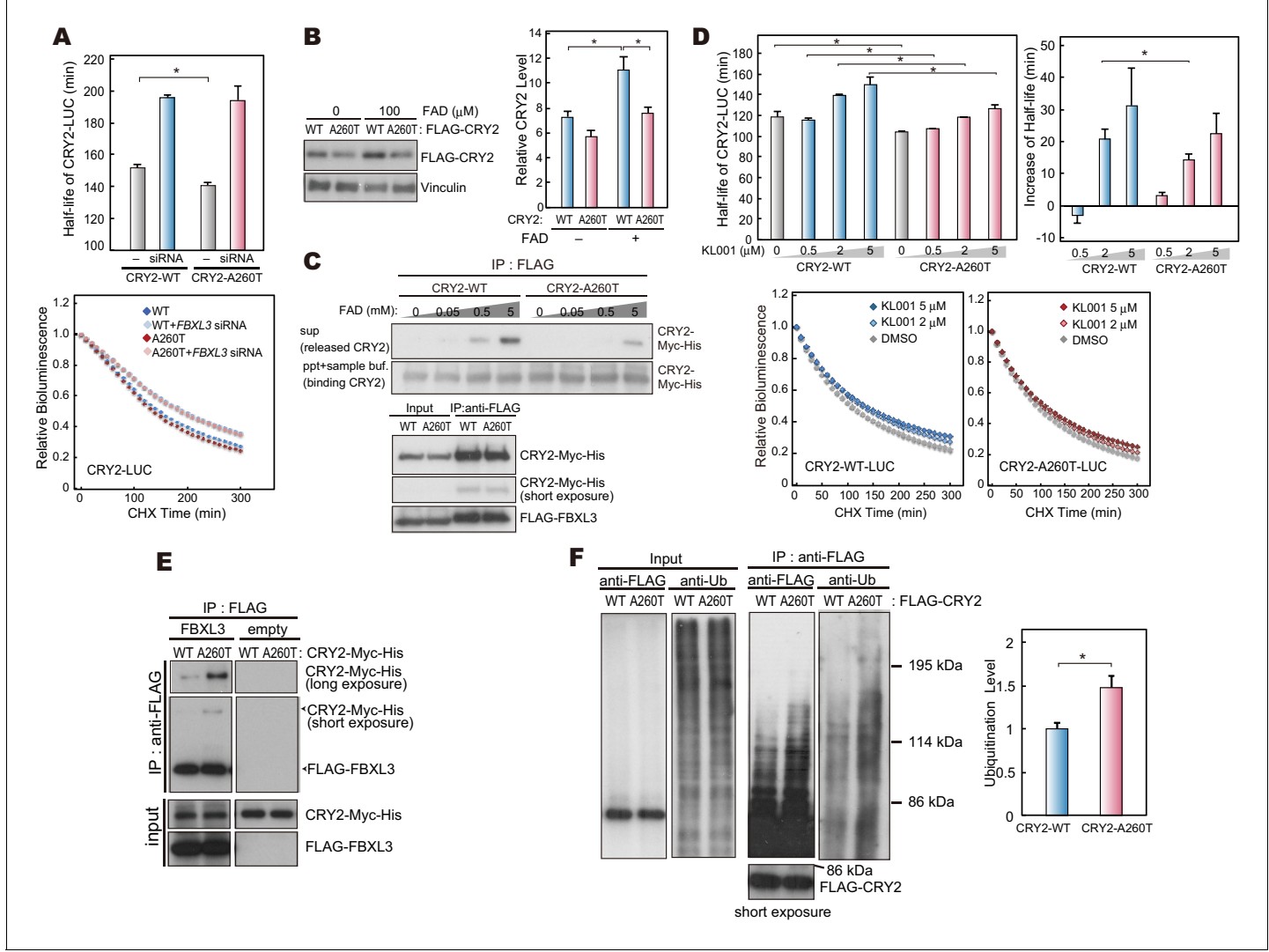

**Figure 5.** CRY2-A260T binds more strongly to FBXL3 leading to faster degradation of mutant CRY2. (**A**) Effect of human *FBXL3* knockdown on CRY2 protein stability in HEK293 cells. Forty-eight hours after transfection, the culture medium was changed to the recording medium containing 100 µg/ml CHX, and bioluminescence of CRY2-LUC was recorded continuously. Bioluminescence normalized to the value at time 0 was fitted to an exponential curve to determine half-life of CRY2-LUC. Data are shown as means with SEM (n = 4, *p<0.05 by Welch's *t*-test). (**B**) Effect of FAD on CRY2 protein levels. Forty-two hours after transfection, HEK293 cells were treated with 100 µM FAD for 6 hr. Data are shown as means with SEM (n = 3, *p<0.05 by Student's *t*-test). (**C**) FAD and FBXL3 competition assay. CRY2-FBXL3 complex expressed in HEK293 cells was purified using FLAG antibody. FAD was added to CRY2-FBXL3 complex and incubated at 4°C for 2 hr in vitro. (**D**) CRY2 protein stability in the cells treated with KL001. Twenty-four hours after transfection, the culture medium was changed to the recording medium containing 100 µg/ml CHX and KL001. Recording of bioluminescence and calculation of half-life was performed as described above. *p<0.05 by Student's *t*-test (n = 3). (**E**) Interaction of CRY2 with FBXL3 in HEK293 cells treated with MG132 for 6 hr prior to harvesting. (**F**) Ubiquitylation of CRY2. HEK293 cells expressing FLAG-CRY2 were treated with MG132 for 6 hr before harvesting. FLAG-CRY2 was then purified and blotted with anti-Ubiquitin antibody. Quantitative data are shown as means with SEM (n = 3, *p<0.05 by Student's *t*-test).

The following figure supplement is available for figure 5:

**Figure supplement 1.** Structural modeling of A260T CRY2.

A260D mutation (*Figure 4—figure supplement 1*). Taken together, we demonstrated that the A260T mutation in the FAD binding pocket endows mutant CRY2 with a higher accessibility and affinity for FBXL3, therefore leading to faster proteasomal degradation of mutant vs. wild-type CRY2.

## Decreased endogenous CRY2 protein levels in h*CRY2-A260T* mice

Our in vitro studies demonstrate that CRY2-A260T is less stable than CRY2-WT (*Figure 4*). To examine the protein levels of mutant vs. wild-type CRY2 under physiological conditions, we used MEFs prepared from transgenic mice. Total protein levels of endogenous CRY2 were significantly lower in synchronized cells derived from h*CRY2-A260T* vs. h*CRY2-WT* mice at two different time points (*Figure 6A*), even though *Cry2* transcript levels in h*CRY2-A260T* mice was higher than in h*CRY2-WT* mice (*Figure 6B*). Similarly, CRY2 protein levels from liver nuclear extracts were lower in h*CRY2-A260T* vs. h*CRY2-WT* mice (*Figure 6C,D*), while h*CRY2* mRNA levels were higher in mutant mice (*Figure 6E*). CRY1 protein levels were also decreased by the mutation in nuclear extracts from liver (*Figure 6C,D*), suggesting that CRY1 is destabilized in h*CRY2-A260T* mice. At the same time, we found that nuclear expression of PER1 and PER2 were up-regulated in liver extracts from mutant mice, particularly at ZT14 (*Figure 6C,D*), suggesting that the timing of nuclear accumulation of PER proteins is advanced in h*CRY2-A260T* mice. Although PER1 and PER2 protein levels were significantly altered, their mRNA levels were not different (*Figure 6E*). A similar alteration of PER1 and PER2 protein levels in the absence of noticeable changes in mRNA levels was previously reported for *Psttm* mutant mouse liver (*Yoo et al., 2013*). *Psttm* mice have a mutation in the *Fbxl21* gene and this mutation decreases the protein level of FBXL21, which functionally competes with FBXL3. The *Psttm* mutation resulted in CRY1 and CRY2 protein destabilization and a shorter circadian period in mice (*Yoo et al., 2013*), which parallels the phenotype of our h*CRY2-A260T* mice (*Figure 2E*). We found that the expression of clock genes in liver was not significantly altered by the mutation, while the effect was obvious in MEFs (*Figure 6B,E*, and *Figure 6—figure supplement 1*). This is congruent with a previous suggestion that the effect of the CRY2 destabilization has diverged among different tissues (*Hirano et al., 2013*; *Yoo et al., 2013*). Collectively, we demonstrated that the A260T mutation destabilizes CRY2 proteins in vivo likely through alteration of FBXL3-mediated CRY2 degradation, leading to perturbation of the circadian clock.

## Discussion

We report here a mutation in h*CRY2* that causes FASP in humans. We initially identified this as a novel variant. Since that time, it has been recognized as a rare variant in the SNP database (rs201220841). The frequency of the A260T allele (0.00008 in the ExAc database) is much lower than that of FASP (0.5%, our unpublished data) in the general population. This is consistent with the A260T variant found in one of our FASP families being responsible for a small portion of FASP in the general population.

Among the mutation carriers of this family (*Figure 1A*), the proband, her twin sister, and her mother have clear advanced sleep phase. In contrast, the nephew of the proband (101374) did not have early sleep onset and offset although his genotype is A/G (*Figure 1—source data 1*). Considering his age, we classified him as 'unknown', since adolescents and young adults are typically more difficult to categorize as having a definite circadian phenotype due to normal phase delays seen in many people beginning in adolescence and persisting into young adult life (boys/men > than girls/ women) (*Roenneberg et al., 2004*). When phenotyped at age 21, subject 101374 was at the statistical peak age of maximum phase delay due to these developmental effects and was prone to be even more phase-delayed by his male sex. Therefore, he may become progressively more phase advanced as he grows older as a result of the *CRY2* FASP allele as it is unmasked by these developmental changes. It is also possible that the mutation may not have 100% penetrance and therefore the nephew will never manifest the FASP trait.

In order to confirm that the A260T mutation is causative of FASP, we generated mice carrying the mutation and subjected them to detailed behavioral analysis (*Figure 2*). Consistent with the other FASP mutations previously reported in h*CK1δ* and h*PER3*, the effects of the human mutation observed in mouse models and in vitro are subtle compared to those found in forward mutagenesis screens. This is expected since the mutations that we identified are found in extant humans in the 'real world'. The circadian body clock plays crucial roles in maintaining normal physiological functions. Thus, any mutation manifesting the strong phenotypes seen in mutagenesis screens would almost certainly have been a selective disadvantage if they arose spontaneously in humans. Furthermore, while the phase advance in mice carrying the human *PER2* mutation appears to be due largely to a shortening of circadian period, the published *PER3* mice and the *CRY2* mice reported here both

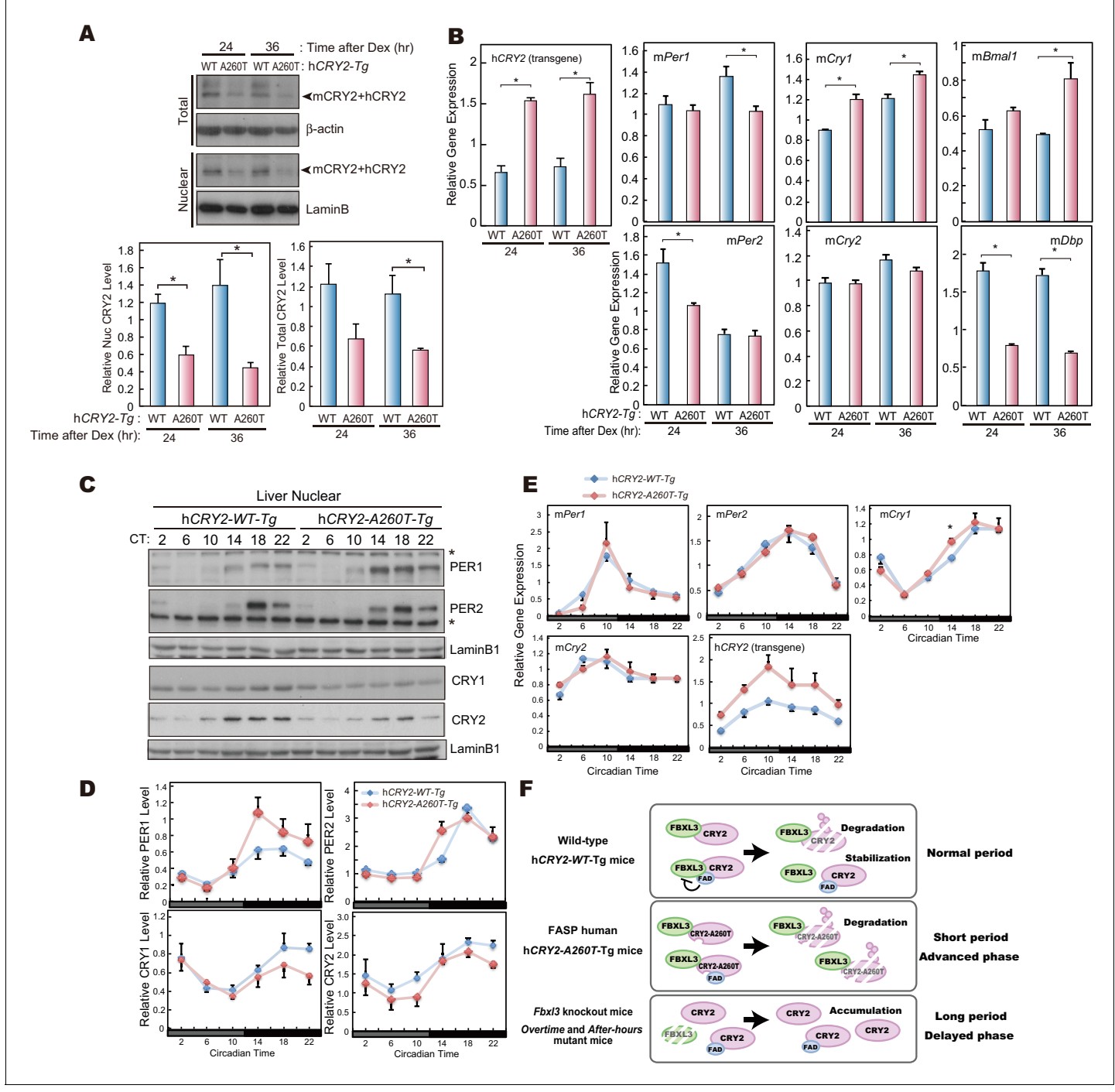

**Figure 6.** CRY2-A260T expression is down-regulated in h*CRY2-A260T* mice. (**A**) CRY2 protein levels in synchronized MEFs. Cells were treated with 100 nM Dex for 2 hr to synchronize the cellular rhythms. Media was change and MEFs were cultured for 24 or 36 hr before harvesting. Quantified band intensities of CRY2 (mouse CRY2 and human CRY2) are shown as means ± SEM (n = 3, *p<0.05 by Student's *t*-test). (**B**) mRNA levels of clock genes in synchronized MEFs. Cellular rhythms of MEFs were synchronized with 100 nM Dex for 2 hr. mRNA levels were quantified by real-time PCR. Data are shown as means ± SEM (n = 3, *p<0.05 by Student's *t*-test). (**C**) Temporal expression profiles of PER1, PER2, CRY1 and CRY2 in mouse liver. Mice were sacrificed every 4 hr on the second day in DD. Asterisks mark non-specific bands. (**D**) Quantification of protein levels in (**C**). Data are shown as means ± SEM (n = 3). (**E**) mRNA levels of indicated clock genes in mouse liver. Mice were sacrificed every 4 hr on the second day in DD. mRNA levels of indicated genes were quantified by real-time PCR using gene specific primers. Data are shown as means ± SEM (n = 3). (**F**) Model of CRY2 protein regulation. In wild-type, FAD binding to CRY2 acts to stabilize by competing with FBXL3. In h*CRY2-A260T* transgenic mice or FASP human subjects with CRY2 mutations, FAD does not protect CRY2 from FBXL3-mediated degradation. Destabilization of CRY2 results in shortened period, leading to

*Figure 6 continued on next page*

*Figure 6 continued*

advanced sleep phase. In *Fbxl3* knockout mice or mutant mice (*Overtime* and *After-hours*) (*Godinho et al., 2007*; *Siepka et al., 2007*), CRY2 is stabilized in the nucleus, thus lengthening the circadian period.
The following figure supplement is available for figure 6:

**Figure supplement 1.** mRNA levels of clock genes in MEF and liver of h*CRY2* BAC Tg mice.

have altered entrainment. We speculate that some of FASP in humans is caused by altered entrainment properties that would not have been detected in forward mutagenesis screens because they have focused almost entirely on measuring period (not phase) as the target phenotype.

Previously, we have used wheel-running behavior analysis to characterize mouse models carrying human FASP mutations (*Xu et al., 2007*, *2005*). Here, we investigated an additional behavior analysis method. For the h*CRY2-A260T* mouse model, we also employed continuous video recording (*Figure 2A* and *Figure 2—figure supplement 1*). We found that video recording was quite sensitive to detect advanced sleep phase (*Figure 2A–C* and *Figure 2—figure supplement 1*). Although wheel-running also displays the phase advance of activity offset in the h*CRY2-A260T* mice (*Figure 2—figure supplement 2*), the data from wheel-running is less robust for detecting a phase advance of activity onset. This is likely due to a strong light-masking effect for mouse at the light-to-dark transition for activity onset. Thus, when the lights are on, mice are less likely to run on a wheel. However, smaller amplitude movements like grooming behavior and moving in the cage were detected by video recording. Wheel running was quite sensitive to detect phase advance of activity offset as this is seen during the dark phase of LD 12:12.

In this study, we found that the circadian period was significantly shortened by the A260T mutation in the central and peripheral clocks (*Figure 2F*, *3A–E*). Growing evidence indicates that stability of CRY proteins dominantly determines the circadian period length (*Godinho et al., 2007*; *Hirano et al., 2014*; *Shi et al., 2013*; *Siepka et al., 2007*; *St John et al., 2014*): stabilization of CRY lengthens circadian period in mice whereas destabilization of CRY shortens the period. Consistent with this model, our human mutation destabilizes CRY2 and leads to a short circadian period of mouse behavioral rhythms (*Figure 6F*). Shortened circadian periods have been measured in one FASP human subject (*Jones et al., 1999*), and in mouse models of human FASP mutations (*Xu et al., 2007*, *2005*). Mutant animals having a shorter free-running period such as *tau* mutant hamster also tend to exhibit advanced phase of behavioral rhythms in LD (*Lowrey et al., 2000*). Thus, it is likely that the *CRY2* mutation results in FASP, at least in part, through shortening of the free-running period.

Interestingly, our data also indicates that another circadian clock feature, phase resetting by light, is dysregulated in the h*CRY2* mutant mouse model. h*CRY2-A260T* mice have a smaller phase-shift in response to a light pulse in early subjective night compared to h*CRY2-WT* mice (*Figure 2E*; *Figure 2—figure supplement 3*). In order to live on a 24 hr day in LD 12:12, wild-type mice need to phase delay a small amount each day because the endogenous circadian period is slightly shorter than 24 hr. A reduced ability to phase-delay observed in the h*CRY2-A260T* mouse model could contribute to advanced sleep phase, though the mechanism for the altered phase-shift remains to be elucidated. Although the difference in the free-running period of the transgenic mice was subtle as compared to the degree of phase-advance manifested in human mutation carriers or mice carrying the mutant h*PER2* transgene (*Figure 2B,F*), it is reasonable to expect that the alteration of both circadian period and light-induced phase-shifting together will strongly influence the phase angle of entrainment. The *after-hours* mutation in the mouse *Fbxl3* gene causes stabilization of CRY leading to an extremely long circadian period (*Godinho et al., 2007*). These mutant mice also exhibit large phase-shifts in response to light (*Guilding et al., 2013*). The authors speculated that reduced amplitude of the circadian clock in *after-hours* mutant mice leads to the abnormal enhancement of phase-resetting. In this study, the A260T mutation elevated PER1 and PER2 levels, especially at ZT14, and the amplitude of PER1 protein rhythm was greater in h*CRY2-A260T* vs. h*CRY2-WT* mice (*Figure 6D*). Thus, the effect of light pulses in the early night may be decreased by the perturbed protein profiles (increased amplitude) of PER1 and PER2 in h*CRY2-A260T* mice.

FAD is a chromophore binding to flavo proteins regulating various biological processes and it is required for the light-sensing activity of CRY in various species (*Lin and Todo, 2005*; *Partch and Sancar, 2005*). *Drosophila* CRY is degraded by the proteasome in response to light signals, which is a trigger for phase-resetting of the circadian clock in flies. However, the ability of mammalian CRYs as a photoreceptor remains controversial as double knockout mice of *Cry1* and *Cry2* are still able to entrain to light and show *Per1* gene induction in SCN in response to light pulses (*Okamura et al., 1999*). These double knockout mice completely lack behavioral rhythms in constant darkness (*van der Horst et al., 1999*). Furthermore, the repressor activity of CRY on E-boxes and its interaction with other clock proteins are independent of light (*Griffin et al., 1999*). These findings emphasize the light-independent role of CRY proteins in mammals. Thus, the physiological role of FAD binding in the mammalian clock has been totally unknown, while a previous study implied that FAD can structurally compete with FBXL3. Our findings provide the first evidence that FAD functions as a stabilizer of CRY2 protein by modulating FBXL3-CRY2 interaction (*Figure 5*). The results presented here demonstrate that the protein stability of CRY2 regulated by the balance of FBXL3 and FAD controls clock speed and sleep/wake timing in mice and humans.

Several genetic studies reported that the human *CRY2* gene is associated with mood regulation, cancer and glucose homeostasis (*Dupuis et al., 2010*; *Hoffman et al., 2010*; *Kovanen et al., 2013*; *Lavebratt et al., 2010*; *Sjöholm et al., 2010*; *Zhang et al., 2013*). Although psychiatric disorders, cancer, and metabolic disorders are tightly connected with dysfunction of the biological clock, associations of *CRY2* polymorphisms with morning/evening preference or other circadian phenotypes have not been described. One polymorphism in the *FBXL3* gene was reported to be associated with diurnal preference (*Parsons et al., 2014*), implying the conserved role of FBXL3 in the human circadian clock. However, it remains to be elucidated whether that variant in *FBXL3* is causative (vs. merely be associated with) the human circadian phenotype and whether it acts through CRY regulation. Here, we demonstrate that regulatory mechanisms for CRY2 protein are well conserved between mice and humans and that control of CRY2 stability is critical for appropriate phase angle and period of the circadian clock in humans.

## Material and methods

### Method summary

All human subjects signed a consent form approved by the Institutional Review Boards at the University of Utah and the University of California, San Francisco (IRB# 10–03952). The consent form includes all confidentiality and ethic guidelines and also indicates not revealing subject information in the publication. All experimental protocols (Protocol no. AN111686-02) were conducted according to US National Institutes of Health guidelines for animal research and were approved by the Institutional Animal Care and Use Committee at the University of California, San Francisco.

### Human data and mutation screening

Subjects were characterized by a previously published procedure established by one of the authors (CRJ, *Jones et al., 1999*). The data were interpreted by one of the authors (CRJ) as possible, probable, definite, or severe advanced sleep phase syndrome by at least age 30. Though ancillary features of ASP (earlier spontaneous wake time if an earlier bed time is selected) and potential confounding or masking influences were considered, most participants categorized as 'definite ASP' reported spontaneous vacation sleep onset and offset time no later than 21:30 and 05:30, respectively and had H-O score (or numerically equivalent childhood MEQ score) of at least 72. We considered children and adolescents more difficult to categorize as having a definite, life-long circadian phenotype unless it was severe by all measures including DLMO phase. DNAs purified from blood samples were used to screen for mutations.

The salivary dim light melatonin onset (DLMO) of the proband was obtained on the last night of the home recordings. DLMO phase was assessed from serial saliva samples (~1 mL) collected at 30 min intervals using 'Salivette' saliva collection tubes (Sarstedt, Inc., Newton, NC) in dim light (</=10 lux) confirmed by recording the ambient light level before each sample using a luxmeter (Sinometer, ShenZhen, China). Samples were collected, beginning 6 hr before the subject's typical bedtime. Saliva samples were frozen overnight and then shipped the next day in an insulated box with frozen

coolant to another laboratory (SolidPhase Inc., Portland, ME) for radioimmunoassay by test kit (ALPCO Diagnostics, Windham, NH). The lower limit of detection of this assay is 0.2 pg/mL. The salivary dim light melatonin onset (DLMO) in adults was calculated and compared with a population sample not purposely selected for morning or evening preference by the method and data of Burgess and Fogg (*Burgess and Fogg, 2008*). Concurrent sleep logs and Zeo (Zeo Incorporated, Boston, MA) EEG recordings were obtained for ten consecutive nights of sleep at home. DNAs purified from blood samples were used to screen for mutations. For this particular family, a list of candidate genes including *CLOCK, BMAL1, PER1-3, CRY1-2, DEC1-2, CSNK1D, CSNK1E, PRKAA2, NPAS2, CSNK2A2, CSNK2B, FBXL3, GSK3B, PKCA, PRKAA1, PRKAA2, RAB3A, RORA, TIMELESS, NR1D1, and PRKCG* were screened. CRY2 (Accession number; EAW68030) A260T was identified as a novel variant specific for mutation carriers of this family (at the time of identification in 2008). The prevalence of the A260T allele (rs201220841) is 0.008% and 0.1% in the two sets of public genome databases, of which sample sizes are 121,412 and 1323, respectively.

## Engineering of BAC constructs for generating transgenic mice

A human BAC RP11-1084E2 containing the entire CRY2 gene on a 189 kb genomic insert was obtained from CHORI (Children's Hospital Oakland Research Institute). The BAC clone was modified by homologous recombination using the Counter-Selection BAC Modification Kit (Gene Bridges GmbH, Heidelberg, Germany) as previously described (*Lee et al., 2012*). Briefly, a linear PCR fragment containing a streptomycin/kanamycin counter selection gene was amplified. The primers for this reaction were designed so that 20 nucleotides would anneal to the streptomycin/kanamycin gene and an additional 40 nucleotides homologous to sequences flanking the mutation site. This PCR product was transferred into the RP11-1084E2 BAC to initiate homologous recombination in the DH10B *Escherichia coli* strain that already contained the plasmid pSC101- BAD-gbaA$^{tet}$. The counter selection gene was then removed by a second recombination event using an oligonucleotide carrying the mutation (G-to-A) in the center. All relevant segments generated by PCR and recombination were sequence confirmed. Detailed mapping was carried out for the modified BACs to ensure that correct constructs were obtained. Transgenic mice were generated using standard microinjection procedures. The transgenic founders were on a C57BL/6 × SJL F$_1$ background and were backcrossed to C57BL/6 mice in successive generations. The copy number for each transgenic line was calculated by quantitative real-time PCR using common sequences for mouse *Cry2* (reference) and human *CRY2* genes.

## Purchased mouse lines

m*Per2$^{Luc}$* knockin mice (RRID IMSR_JAX006852) and m*Cry2* knockout mouse (RRID IMSR_JAX016185) were purchased from The Jackson Laboratory and crossed with h*CRY2* transgenic mice.

## Wheel-running analysis of transgenic mice

All mice tested were ~8 week-old males maintained on a C57BL/6J background. Mice were kept in individual wheel running cages with free access to food and water. First, mice were entrained to LD 12:12. Activity profiles, offset time and acrophase were analyzed using data from day 10 to day 14 in LD. After entrainment to LD for approximately 3 weeks, mice were released into constant darkness (DD) for measurement of free-running period. Circadian periods were calculated by line fitting of activity onsets from day 7 to day 19 in DD. To analyze for phase-shifts, mice were given a 30 min-light pulse (200 lux) beginning at ZT14 (2 hr after lights-off) or at ZT22 (2 hr before lights-on), and then released into DD. Phase-shifts were determined by line fitting of activity onsets from day1 to day7 in DD. All data collection and analysis was done using ClockLab software (Actimetrics, Wilmette, IL; RRID SCR_014309). Activity onset and offset were defined using the ClockLab software algorithm. The default template is 6 hr of inactivity followed by 6 hr activity for onset (or vice versa for offset).

## ANY-maze analysis of transgenic mice

All mice tested were ~16 week-old males maintained on a C57BL/6J background. Mice were kept in individual cages with free access to food and water. Mice were monitored by infrared camera and tracked by an automatic video tracking system (Stoelting Co., Wood Dale, IL; RRID SCR_014289).

Mice were entrained to LD 12:12 for 1 week and then locomotor activity was recorded for 3 or 4 days. Walking distance and immobility times were calculated using ANY-maze software and data were averaged. Samples with over 500-meter walking distance or below 10,000-s immobility time each day were excluded from the statistical analysis due to the failure of automatic tracking.

## Cell culture and constructs

HEK293 cells (ATCC CRL-1573; RRID CVCL_0045) and NIH 3T3 cells (ATCC CRL-1658; RRID CVCL_0594) were purchased from ATTC. Authentication of the cell lines was performed using STR profiling by ATCC. Mycoplasma contamination was checked every 6 months and mycoplasma-free cell lines were used for all experiments in this study. Cells were cultured in DMEM (Sigma Aldrich, St. Louis, MO) containing 10% FBS and 100 U/ml Penicillin-Streptomycin (Thermo Fisher Scientific, South San Francisco, CA) and maintained by standard methods. Mouse embryonic fibroblasts (MEFs) were prepared from E12.5 embryos of h*CRY2-WT* and h*CRY2-A260T* transgenic mice. After removing the head, paws and internal organs, embryos were chopped and incubated in 0.25% trypsin in PBS for 24 hr at 4°C. After incubation for 20 min at 37°C in 0.25% trypsin in PBS, cells were dissociated by pipetting in DMEM. Supernatant was cultured in a cell culture dish with DMEM and maintained by standard methods. Cells were transfected with Lipofectamine 3000 transfection reagent (Thermo Fisher Scientific) according to manufacturer's protocol. DNA constructs used for transfections are as follows: hCRY2-WT-Myc-His/pcDNA3.1, hCRY2-A260T-Myc-His/pcDNA3.1, hCRY2-A260D-Myc-His/pcDNA3.1, hCRY2-WT-HA/pCMV-tag2B, hCRY2-A260T-HA/pCMV-tag2B, FLAG-hCRY2-WT/p3×FLAG-CMV-10, FLAG-hCRY2-A260T/p3×FLAG-CMV-10, FLAG-hCRY1-WT/p3×FLAG-CMV-10, FLAG-hCRY1-A241T/p3×FLAG-CMV-10, FLAG-hFBXL3/p3×FLAG-CMV-10, FLAG-hBMAL1/p3×FLAG-CMV-10, FLAG-hCLOCK/p3×FLAG-CMV-10, hCRY2-WT-LUC/p3×FLAG-CMV-10, hCRY2-A260T-LUC/p3×FLAG-CMV-10, hCRY2-A260D-LUC/p3×FLAG-CMV-10, m*Per1-luc*/pGL3, pRL-TK (Renilla luc expression for internal control in luciferase assay, Promega, Fitchburg, WI). 0.3kb-m*Bmal1*-luc/pGL3 is a gift by Dr.Yoshitaka Fukada (University of Tokyo). Mutant hCRY2 and hCRY1 expression vectors were generated by PCR-based site-directed mutagenesis, and the mutation was verified by sequencing. For knockdown of human *FBXL3*, Hs_FBXL3_1, Hs_FBXL3_2 FlexiTube siRNA (QIAGEN, Hilden, Germany) and control siRNA (QIAGEN) were purchased.

## Bioluminescence rhythms in tissue culture

h*CRY2* transgenic mice were crossed with *mPer2^Luc* knock-in mice (*Yoo et al., 2004*; RRID IMSR_JAX006852). Mice were sacrificed between ZT11 and ZT12. Dissected liver tissues were cultured on Millcell culture membrane (PICMORG50, EMD Millipore, Billerica, MA) in 35 mm dishes. For recording of lung rhythms, dissected lung tissue was placed in 35 mm dishes without Millcell culture membrane. Recording medium was phenol-red free DMEM (Sigma Aldrich) containing 10 mM HEPES-pH7.0, 3.5 g/L D-glucose, 0.2 mM luciferin potassium salt, 0.35 g/L sodium bicarbonate, 2% B-27 supplement (Thermo Fisher Scientific), 50 U/ml penicillin-streptomycin (Thermo Fisher Scientific). Bioluminescence was continuously recorded in a LumiCycle 32 instrument (Actimetrics, Wilmette, IL). Bioluminescence was detrended by subtracting 24 hr average of bioluminescence using the LumiCycle analysis software. The periods were determined by dampened sine-curve fitting using LumiCycle analysis.

## Bioluminescence rhythms in cell culture

For h*CRY2* transgenic/m*Cry2* knockout MEFs and NIH3T3 stable cells, cells were transfected with 500 ng 0.3kbp-m*Bmal1*-luc/pGL3 by Lipofectamine3000 before recordings. Cellular rhythms were synchronized by treatment with 100 nM dexamethasone (DEX) for 2 hr. Medium was changed to the recording medium: phenol-red free DMEM (Sigma Aldrich) containing 10 mM HEPES-pH7.0, 3.5 g/L D-glucose, 0.1 mM luciferin potassium salt and 50 U/ml penicillin-streptomycin (Thermo Fisher Scientific). Bioluminescence recording and data analysis were as described in the methods for "Bioluminescence rhythms in tissue culture".

## Luciferase assay

HEK293 cells were transfected with 50 ng *Per1*-luc expression vector, 25 ng Renilla luc control vector and 2, 5, 10 or 20 ng hCRY2 expression vectors. The luciferase assay was performed with Dual-

Luciferase Reporter Assay System (Promega, Fitchburg, WI) according to the manufacturer's protocol. Bioluminescence was detected by Synergy H4 Hybrid Multi-Mode Microplate Reader (BioTek, Winooski, VT). Bioluminescence of Firefly LUC was normalized to bioluminescence of Renilla LUC.

## Luciferase-based degradation assay

The hCRY2-LUC fusion protein expressing vector was created by inserting a *CRY2-Luc* cDNA between EcoRI and BamHI sites in the p3×FLAG-CMV-10 vector. HEK293 cells were transfected with 50 ng hCRY2-LUC vectors and cultured for 24 hr. The culture medium was replaced with recording medium [phenol-red free DMEM (Sigma Aldrich) supplemented with 10% fetal bovine serum, 3.5 mg/ml glucose, 50 U/ml penicillin-streptomycin (Thermo Fisher Scientific), 0.05 mM luciferin, and 10 mM HEPES-NaOH; pH 7.0] containing 100 µg/ml cycloheximide (CHX; Santa Cruz Biotechnology Inc., Santa Cruz, CA). Luciferase activity of hCRY2-LUC was recorded at 10-min intervals at 37°C with a LumiCycle 32 instrument (Actimetrics). The luminescence signals were fitted to an exponential function to quantify the half-life of CRY2-LUC. KL001 (Cayman Chemical, Ann Arbor, MI) was diluted in DMSO to a final concentration of 20 mM.

## Expression profiles of proteins and genes

Mice were entrained to LD 12:12 for at least 10 days. Mice were transferred to DD, and mice were sacrificed in dim red light on the 2nd day of DD. Liver tissues were collected, followed by nuclear extraction (*Yoshitane et al., 2009*) and mRNA extraction. Protein levels and mRNA levels were normalized to LaminB levels and *Gapdh* levels, respectively.

## Western blotting

For whole-cell extracts, HEK293 cells were lysed in SDS sample buffer [62.5 mM Tris-HCl (pH 6.8), 50 mM DTT, 2% SDS, 10% glycerol]. Preparation of the cytosolic and the nuclear fractions of mouse liver was performed as previously described (*Yoshitane et al., 2009*). Protein samples were separated by SDS-PAGE. Tissues were transfered to PVDF membranes (EMD Millipore) with blocking in T-TBS [50 mM Tris-HCl (pH 7.4), 137 mM NaCl, 0.1% Tween 20] containing 1% Skim milk. Primary antibodies were reacted in the blocking solution at 4°C overnight. Then, secondary antibodies were reacted in the blocking solution at RT for 2 hr. Proteins were detected with the Western Lightning Plus ECL (PerkinElmer, Waltham, MA). Band intensities were determined using Image J software. β-actin and Vinculin were used as loading controls for total cell lysates, and LaminB and TBP were used as nuclear markers. Proteins were detected with the following antibodies: anti-cMyc 9E10 (Santa Cruz, sc-40), anti-FLAG M2 (Sigma Aldrich, F1804), anti-HA Y11 (Santa Cruz, sc-805-G), anti-β-actin (Abcam, AC-15, Cambridge, UK), anti-Vinculin (Abcam, ab18058), anti-TBP (Santa Cruz, sc-273), anti-Ub (Santa Cruz, sc-8017), anti-hPER1 (Thermo Fisher Scientific, PA1-524), anti-LaminB1 (Abcam, ab16048 and Santa Cruz, C20), anti-mPER2 (Alpha Diagnostic International, PER-21A, San Antonio, TX), anti-hCRY2 (Santa Cruz, sc-130731) and anti-mCRY1 (MBL, PM081, Woburn, MA). Rabbit polyclonal anti-mCRY2 antibody was provide by Dr.Yoshitaka Fukada (University of Tokyo) (*Hirano et al., 2013*). Secondary antibodies used were goat anti-mouse IgG-HRP (Santa Cruz, sc-2005), goat anti-rabbit IgG-HRP (Santa Cruz, sc-2006) and goat anti-guinea pig IgG-HRP (Santa Cruz, sc2438).

## FAD competition assay

Flavin adenine dinucleotide disodium salt hydrate (FAD, Sigma Aldrich) was diluted in PBS to a final concentration of 100 mM. HEK293 cells were transfected with plasmid vectors for 10 µg hCRY2-His-Myc (WT or A260T) and 10 µg FLAG-hFBXL3. Forty-two hours after transfection, the cells were treated with 10 µM MG132 (EMD Millipore) for 6 hr. CRY2-FBXL3 complex was purified with anti-FLAG M2 affinity gel (Sigma Aldrich). FAD was incubated with CRY2-FBXL3 complex binding to anti-FLAG M2 affinity gel in 40 µl PBS for 2 hr on ice. After centrifugation, the supernatant was collected as the 'released CRY2' sample. CRY2 still binding to FLAG-FBXL3 was eluted by adding SDS sample buffer to FLAG-M2 affinity gel.

## Immunoprecipitation

HEK293 cells were transfected with plasmid vectors for hCRY2-His-Myc (WT or A260T) and FLAG-hFBXL3. Forty-two hours after transfection, the cells were treated with 10 µM MG132 (EMD

Millipore) for 6 hr. CRY2-FBXL3 complex was purified with anti-FLAG M2 affinity gel (Sigma Aldrich) and eluted by 300 µg/ml 3×FLAG peptide (Sigma Aldrich).

## Real-time qPCR

Total RNA was extracted by TRIzol reagent (Thermo Fisher Scientific) from MEFs or liver samples of transgenic animals. cDNA was synthesized by Superscript III (Thermo Fisher Scientific) for MEFs or GoScript (Promega) for liver samples. Quantification of mRNA was performed with GoTaq Real-Time qPCR Kits (Promega) using gene specific primers. mRNA levels were normalized to mouse *Gapdh* levels.

Primers: mouse *Per1*-fw; CAGGCTAACCAGGAATATTACCAGC,
mouse *Per1*-rv; CACAGCCACAGAGAAGGTGTCCTGG,
mouse *Per2*-fw; ATGCTCGCCATCCACAAGA,
mouse *Per2*-rv; GCGGAATCGAATGGGAGAAT,
mouse *Gapdh*-fw; ACGGGAAGCTCACTGGCATGGCCTT,
mouse *Gapdh*-rv; CATGAGGTCCACCACCCTGTTGCTG,
mouse *Cry2*-fw; GGGACTCTGTCTATTGGCATCTG,
mouse *Cry2*-rv; GTCACTCTAGCCCGCTTGGT,
mouse *Cry1*-fw:CCCAGGCTTTTCAAGGAATGGAACA
mouse *Cry1*-rv:TCTCATCATGGTCATCAGACAGAGG
human *CRY2*-fw; CCAAGAGGGAAGGGCAGGGTAGAG,
human *CRY2*-rv; AGGATTTGAGGCACTGTTCCGAGG
mouse *Dbp* FW,AATGACCTTTGAACCTGATCCCGCT
mouse *Dbp* RV,GCTCCAGTACTTCTCATCCTTCTGT
mouse *Bmal1* FW,GCAGTGCCACTGACTACCAAGA
mouse *Bmal1* RV,TCCTGGACATTGCATTGCAT
mouse *Rev-erbα* FW,GGGCACAAGCAACATTACCA
mouse *Rev-erbα* RV,CACGTCCCCACACACCTTAC
mouse *REV-erbβ* FW,TGGGACTTTTGAGGTTTTAATGG
mouse *REV-erbβ* RV,GTGACAGTCCGTTCCTTTGC
mouse *Dec1* FW,ATCAGCCTCCTTTTTGCCTTC
mouse *Dec1* RV,AGCATTTCTCCAGCATAGGCAG
mouse *Dec2* FW,ATTGCTTTACAGAATGGGGAGCG
mouse *Dec2* RV,AAAGCGCGCGAGGTATTGCAAGAC

## Structural modeling

Structural modeling was based on the structure of mouse CRY2 bound to FAD (PDB code 4I6G) and mouse FBXL3 (PDB code 4I6J) (*Xing et al., 2013*). Modeling of mutant CRY2 was performed using Molecular graphics and analyses were performed with the UCSF Chimera package (RRID SCR_004097). Chimera was developed by the Resource for Biocomputing, Visualization, and Informatics at the University of California, San Francisco (supported by NIGMS P41-GM103311) (*Pettersen et al., 2004*).

## Statistics analysis

All error bars in the figures represent SEM except for *Figure 3C*. In *Figure 3C*, the error bars represent SD. No statistical analysis was used to predetermine the sample sizes. Experiments were not randomized and not analyzed blindly. In *Figure 2A* and *Figure 2—figure supplement 1*, the sample with extremely abnormal walking distance (>500 meter walking distance, due to a failure of automatic video tracking) was excluded as an outlier according to the Smirnov-Grubbs test. Data was statistically analyzed using R software (RRID SCR_001905). To assess statistical significance, data was obtained from three or more independent experiments. All data sets were assumed to follow normal distributions by the Kolmogorov-Smirnov test, and homogeneity of variance between compared groups was tested by F-test (comparison of 2 groups) or Bartlett test (comparison of multiple groups). Two-tailed paired Student's *t*-test or Welch's *t*-test was used for the comparison of 2 groups with or without homogeneity of variance. Tukey's test or Games-Howell test were used for

multiple comparisons with or without of homogeneity of variance. Differences with a p value <0.05 were considered statistically significant.

## Acknowledgements

This work was funded by NIH grant GM079180 and HL059596 to LJP and Y-HF and by the William Bowes Neurogenetics Fund. The initial sequencing and analysis were performed at Lawrence Berkeley National Laboratory and at the United States Department of Energy Joint Genome Institute (Department of Energy Contract DE-AC02-05CH11231, University of California). The authors wish to thank Drs. Philip Kurien and Pei-Ken Hsu and Mr. David Wu for suggestions and critical reading of the manuscript. We thank Dr. Yoshitaka Fukada (University of Tokyo) for providing anti-CRY2 antibody and 0.3 kbp-m*Bmal1*-luc construct. LJP is an investigator of the Howard Hughes Medical Institute. AH was supported by the Japanese Society for the Promotion of Science (JSPS) and the Uehara Memorial Foundation (Japan).

## Additional information

### Competing interests

LJP: Reviewing editor, *eLife*. The other authors declare that no competing interests exist.

### Funding

| Funder | Grant reference number | Author |
| --- | --- | --- |
| Japan Society for the Promotion of Science | | Arisa Hirano |
| Uehara Memorial Foundation | | Arisa Hirano |
| National Heart, Lung, and Blood Institute | HL059596 | Louis J PtÃ¡Äek |
| William K. Bowes, Jr. Foundation | Neurogenetics Fund | Ying-Hui Fu |
| National Institute of General Medical Sciences | GM079180 | Ying-Hui Fu |

The funders had no role in study design, data collection and interpretation, or the decision to submit the work for publication.

### Author contributions

AH, Conception and design, Acquisition of data, Analysis and interpretation of data, Drafting or revising the article; GS, Acquisition of data, Contributed unpublished essential data or reagents; CRJ, AL, LAP, WCH, Acquisition of data, Analysis and interpretation of data; YX, Contributed unpublished essential data or reagents; TM, MY, Acquisition of data; LJP, Y-HF, Conception and design, Analysis and interpretation of data, Drafting or revising the article

### Author ORCIDs

Ying Xu, http://orcid.org/0000-0002-6689-7768
William C Hallows, http://orcid.org/0000-0001-8306-8438
Ying-Hui Fu, http://orcid.org/0000-0002-6628-0266

### Ethics

Human subjects: All human subjects signed a consent form approved by the Institutional Review Boards at the University of Utah and the University of California, San Francisco (IRB# 10-03952). The consent form includes all confidentiality and ethic guidelines and also indicates not revealing subject information in the publication.

Animal experimentation: All experimental protocols (Protocol no. AN111686-02) were conducted according to US National Institutes of Health guidelines for animal research and were approved by the Institutional Animal Care and Use Committee at the University of California, San Francisco.

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
