## [Decision Letter]

Thank you for submitting your article "A Cryptochrome 2 Mutation Yields Advanced Sleep Phase in Human" for consideration by *eLife*. Your article has been reviewed by three peer reviewers, including Chris S Colwell, and the evaluation has been overseen by a Senior Editor and Joseph Gleeson as the Reviewing Editor.

The reviewers have discussed the reviews with one another and the Reviewing Editor has drafted this decision to help you prepare a revised submission.

Summary:

The authors identify a rare mutation in the circadian clock gene Cry2, which they find to segregate with advanced sleep phase in a small pedigree. A mouse model carrying the candidate variant exhibits modest shortening of the circadian period as well as altered phase delay, but not phase advance. At the molecular level, the authors provide evidence for altered protein stability of Cry2 due to the mutation. At the molecular level, the authors demonstrate shorter PER2-driven bioluminescence rhythms in the peripheral oscillators. Importantly, the authors examine the impact of the CRY2 mutation on the normally repressing function of the protein, and demonstrate a possible mechanism through altered interaction with a known binding partner, FBXL3. The results provide insight into the regulation of human sleep timing. However, several of the authors' claims are compromised somewhat by an inconsistent quality in the data. Some revisions are required before a conclusion can be reached on whether all of the authors' claims can be supported.

Essential revisions:

1) The characterization of the sleep and circadian phenotypes for the subjects is scarce. This section should be expanded to include e.g. typical bed- and wake times, midsleep on free days, H-O scores, etc. Currently there are only gives H-O scores for the sibling pair and DLMO for a single individual. Please provide basis for classification into categories on left of Figure 1 for all subjects. Also, legend on left includes a symbol for 'probably affected', yet all subjects in pedigree are colored as either 'affected', 'unaffected' or 'unknown'. Who is 'probably affected'? Also provide genotypes for all subjects. Depending on how many subjects in this fairly small kindred can be definitively classified, segregation of the allele with FASPS phenotype may or may not be consistent. Similarly, dim light melatonin onset for normal subjects at should be reported so that a lay reader can assess the degree of phase advance shown by the proband.

2) The tissue/cell rhythms presented in Figure 3 show a pronounced difference in cycling amplitude between genotypes. This applies especially to lung, which seems to only be borderline rhythmic in the mutant transgenic. Period and amplitude should be determined by appropriate curve-fitting methods rather than selected peak-trough distance and a measure of reliability for both parameters needs to be provided. Similarly, poor rhythms in NIH3T3 cells (Figure 3) make these results hard to interpret. This experiment should be repeated with primary MEFs from the respective transgenic lines as well as Cry2-deficient MEFs (previously published), which have been reconstituted with the respective wild type and mutant Cry2 form.

3) Interaction with Fbxl3: The Fbxl3-FAD competition assay (Figure 5) is impossible to interpret because the amount of Fbxl3 and Cry2 in the purified complex is either not shown or the gel is vastly overexposed. In 5D, it remains unclear whether KL001-mediated Cry2 stabilization is indeed reduced, because of the reduction in Cry2 half-life in the absence of the drug. To assess the effect of KL001, the data should be presented as difference in half-life compared to the no-drug control for each construct. Interaction of Cry2 with Fbxl3 (Figure 5) should be confirmed in a physiologically relevant setting (co-immunoprecipitation from transgenic cells/tissues and/or Cry2-deficient cells reconstituted with either Cry2 form, see previous comment).

4) The CRY2-A260T mice have normal activity onset, which does not recapitulate the phenotype shown by FASP subjects. The degree of phase advance in activity offset is also nowhere near that of the human subjects, and the slight change in period (15-20 min decrease) is also unlikely to cause the same degree of advanced sleep phase.

[Editors' note: further revisions were requested prior to acceptance, as described below.]

Thank you for resubmitting your work entitled "A Cryptochrome 2 Mutation Yields Advanced Sleep Phase in Human" for further consideration at *eLife*. Your revised article has been favorably evaluated by a Senior editor, a Reviewing editor (Joseph Gleeson), and two reviewers.

The manuscript has been improved but there are some remaining issues that need to be addressed before acceptance, as outlined below. The two reviewers have provided specific areas for improvement of the text that *eLife* thinks will improve and further clarify your findings.

Please address these comments in a rebuttal letter and submit a revised manuscript for further consideration.

Reviewer #1:

Although parts of the manuscript are much improved over the previous version, some issues would still benefit from further clarification as follows:

1) Second paragraph of Introduction section: "Mutations in mouse Fbxl3 or knockout of the Fbxl3 gene dramatically lengthens the period of mouse behavioral rhythms in constant darkness (Godinho et al., 2007; Hirano et al., 2013; Shi et al., 2013; Siepka et al., 2007), indicating that the protein stability of CRY2 is a critical determinant of circadian period in mice."

This is a misleading statement – all of the references cited studied the combined effects on CRY1 and CRY2 of Fbxl3. The authors should therefore say: "the protein stability of CRY1 and CRY2 is a critical determinant…".

2) Results section: "No other novel mutations were found in ~25 candidate circadian genes that were sequenced."

This statement is unclear. The Cry2 mutation studied is present in public databases (rs201220841), albeit at low frequency. Are there coding variations in the other sequenced circadian genes? If so, were these discounted based on their presence at a higher frequency in public databases, or by other criteria?

3) [Supplementary-material SD1-data] and Figure 1:

Although the subject IDs are not given in the pedigree, it seems based on the ages that subject 101374 would have to be the pro-band's nephew. In that case, he appears to be a carrier of the allele, although he has very late sleep times and a low H-O score. So, does the allele fail to segregate with the FASP phenotype in this case? At 21 years old, it seems doubtful that age is a sufficient explanation for this discrepancy. Penetrance and age of onset should be discussed.

4) Figure 2—figure supplement 1:

The authors describe a 'larger amplitude advance' on a Cry2-null background, but this is not clear from the figure. Have the results in the histograms been tested for statistical significance (as in Figure 2)?

5) Figure 2; Figure 2—figure supplement 2:

Results on WT background are not quantified, so, the authors' claims on activity onset and offset times are hard to follow based on actograms from a single animal. In the supplementary figure, there appears to be a difference in activity offset in hCRY2-A260T-Tg compared to the hCry2-WT-Tg, but not the wildtype control?

6) Figure 2—figure supplement 4:

Did the authors confirm whether the transgenes in fact rescues the Cry2 phenotype in the KO strain? The WT-Tg still seems to have a long period, is this actually different from the Cry2 KO background strain?

7) Results section, “Shortened period and reduced phase-shift in *hCRY2-A260T* mice”: "Of note, there is no significant difference in the periods and phase-shifting of hCRY2-WT transgenic vs. transgene negative mice (Figure 2), indicating that the shortening of circadian period and abnormal phase-delay are not simply due to overexpression of hCRY2."

This is not the case for the activity offset in Figure 2—figure supplement 2, though?

8) Figure 4—figure supplement 1: The quantification does not seem to match the shown WB at all. Rather, in the WB, the WT and mutant results seem essentially indistinguishable.

*Reviewer #2:*

Overall, I feel that the authors have done a good job addressing the reviewer's concerns and that the manuscript has been significantly strengthened. I agree with the issue that that this is a small kindred with the sleep and circadian phenotypes for the subjects relatively weak. But the mouse work is extensive. Also, I remain concerned that the CRY2-A260T mice have normal activity onset that does not recapitulate the phenotype shown by FASP subjects. Nevertheless, I feel that the video analysis of activity and sleep is a perfectly reasonable way to measure behavioral rhythms. Overall, I continue to feel that this work will make a meaningful contribution to the literature and is appropriate to publish in *eLife*.

---

## [Author Response]

*Essential revisions:*

*1) The characterization of the sleep and circadian phenotypes for the subjects is scarce. This section should be expanded to include e.g. typical bed- and wake times, midsleep on free days, H-O scores, etc. Currently there are only gives H-O scores for the sibling pair and DLMO for a single individual.*

We have added a table summarizing human sleep phenotype in this submission (Figure 1—figure supplement Table. 1). We have interviewed the proband’s mother who also has the A260T mutation. She is not included in the Supplement Table 1. We could not perform sleep log recording or questionnaire diagnosis because she was 91 years old when phenotyped. However, in the interview, she described that she habitually woke up at 5:00 in the morning without alarm clock at least since she was 31 years old when she gave birth to her first child. It was hard to categorize the proband’s nephew because of his young age.

*Please provide basis for classification into categories on left of Figure 1 for all subjects. Also, legend on left includes a symbol for 'probably affected', yet all subjects in pedigree are colored as either 'affected', 'unaffected' or 'unknown'. Who is 'probably affected'? Also provide genotypes for all subjects.*

The information is included in the Methods section. The figure has been corrected, and genotypes been added accordingly.

*Depending on how many subjects in this fairly small kindred can be definitively classified, segregation of the allele with FASPS phenotype may or may not be consistent.*

We understand the family in this study is not big enough to conclude that A260T mutation is causative for FASP base on human genetics data alone (i.e. by linkage analysis). That’s why we generated the mouse model and characterized their behavior to see how A260T mutation affects behavioral and cellular rhythms on a homogeneous genetic background. One thing that we’ve learned from study of familial human behavioral traits is the challenge of recruiting additional family members. Sleep is NOT a disease phenotype, so people are not as motivated to participate with all the questionnaires, interviews, and tests that we would like them to do, especially when they are not getting paid for any of these. In order to overcome this challenge, our approach has been trying our best to recruit as many subjects as possible in each family followed by mutation screening. Proving that a mutation is causal requires demonstration of the behavioral phenotype in a mouse model and the altered molecular functions in vitro/in vivo. If the family has many mutation carriers who have the sleep trait but the mouse model does not demonstrate the phenotype, then we cannot follow up with the mutation even if the number of mutation carriers is higher.

*Similarly, dim light melatonin onset for normal subjects at should be reported so that a lay reader can assess the degree of phase advance shown by the proband.*

We added this information as suggested in the Results of the main text.

*2) The tissue/cell rhythms presented in Figure 3 show a pronounced difference in cycling amplitude between genotypes. This applies especially to lung, which seems to only be borderline rhythmic in the mutant transgenic. Period and amplitude should be determined by appropriate curve-fitting methods rather than selected peak-trough distance and a measure of reliability for both parameters needs to be provided. Similarly, poor rhythms in NIH3T3 cells (Figure 3) make these results hard to interpret. This experiment should be repeated with primary MEFs from the respective transgenic lines as well as Cry2-deficient MEFs (previously published), which have been reconstituted with the respective wild type and mutant Cry2 form.*

We repeated the experiment and analyzed the circadian period by sine-curve fitting for all PER2::LUC data and include this in the revised manuscript. The period was almost the same as those calculated by times of peaks and troughs. There were no significant differences in the amplitudes in lung, liver, and a stable cell line. We replaced with new dataset having similar amplitudes in Figure 3.

We have added data showing the cellular rhythms of Tg mouse MEFs on a WT background (Figure 3) and the Cry2 KO background (Figure 3). The period of MEFs from CRY2-A260T mice was significantly shorter than that of WT Tg mice on both backgrounds.

A measure of reliability (goodness of sine-fitting in Lumicycle Analysis):

Figure 3, Liver

The averaged goodness of sine-fitting was 91.9 ± 3.4% for CRY2-WT and 86.5 ± 3.7% for A260T, respectively.

Figure 3, Lung

The averaged goodness of sine-fitting was 98.2 ± 0.32% for CRY2-WT and 84.8 ± 4.5% for A260T, respectively.

Figure 3, MEF WT background

The averaged goodness of sine-fitting was 89.9 ± 0.43% for CRY2-WT, 93 ± 0.18% for A260T line #1 and 94.2 ± 0.19% for A260T line #2 respectively.

Figure 3, MEF Cry2 KO background

The averaged goodness of sine-fitting was 80.4 ± 1.04% for CRY2-WT Cell line#1, 82.1 ± 2.78% for CRY2-WT Cell line#2, 81.0 ± 0.82%for CRY2-A260T Cell line #1, 78.0 ± 1.63%for CRY2-A260T Cell line #2 and 78.3 ± 1.45%for Cry2 KO, respectively.

Figure 3, Stable cell line, 3T3 cells

The averaged goodness of sine-fitting was 93.4% for CRY2-WT and 96.8% for A260T, respectively.

*3) Interaction with Fbxl3: The Fbxl3-FAD competition assay (Figure 5) is impossible to interpret because the amount of Fbxl3 and Cry2 in the purified complex is either not shown or the gel is vastly overexposed.*

We replaced Figure 5 (current 5C) with a new dataset, showing proper controls (CRY2-FBXL3 complex before adding FAD as well as CRY2 still binding FBXL3 after incubation with FAD) as suggested.

*In 5D, it remains unclear whether KL001-mediated Cry2 stabilization is indeed reduced, because of the reduction in Cry2 half-life in the absence of the drug. To assess the effect of KL001, the data should be presented as difference in half-life compared to the no-drug control for each construct.*

In the initial submission, data was shown for both WT and Mutant without KL001, and WT and Mutant with KL001. We’ve added a new panel to Figure 5 and in Figure 5—figure supplement A. The new panel shows differences in half-life compared to the no-drug control as this reviewer recommended.

*Interaction of Cry2 with Fbxl3 (Figure 5) should be confirmed in a physiologically relevant setting (co-immunoprecipitation from transgenic cells/tissues and/or Cry2-deficient cells reconstituted with either Cry2 form, see previous comment).*

We agree that the change in CRY2 interaction with FBXL3 will need to be confirmed in a physiological condition (i.e. in vivo). However, there is currently no specific antibody for human CRY2 that is good enough to perform co-immunoprecipitation (We in fact have tested all available antibodies for this). This set of experiments will need to be carried out when that reagent becomes available.

*4) The CRY2-A260T mice have normal activity onset, which does not recapitulate the phenotype shown by FASP subjects. The degree of phase advance in activity offset is also nowhere near that of the human subjects, and the slight change in period (15-20 min decrease) is also unlikely to cause the same degree of advanced sleep phase.*

It is well known that mice have strong masking effect by light. Here, it appears that activity onset around the LD transition is masked by light signal in wheel-running analysis, although we detected advanced activity offset. However, Figure 2 shows the advanced “locomotor” activity onset by video recording. Thus, we think that while CRY2-A260T Tg mice wake earlier than WT mice, they are not running on the wheel, likely due to masking to light. In addition, the inconsistency among the analysis tools can be explained by the fact that wheel-running exercise alters mouse behavioral rhythms (Webb et al., Behavioral Neuroscience, 2014) (especially shortens the free-running period). Thus, wheel-running does not reflect the “sleep-wake” pattern perfectly. We also need to remember that humans do not run on wheels all day long, so wheel-running behavior is likely not the best reflection of human sleep wake behavior. We discussed these points and also the differences in the phenotype of our mice in the Discussion section. Also, we understand so little about the relationship between period length and phase angle of entrainment to speak with too much certainty about what is expected in terms of phase advance for any given endogenous period. We accept that there are differences between mouse and human physiology. The conditions under which we measure such things is also different in mice than in humans (see above). Finally, it is also possible that the phase advance in the humans is due to a combination of effects of period shortening and altered entrainment. We have attempted to enroll these subjects in studies in temporal isolation by our collaborator, Jeanne Duffy. Unfortunately, these subjects did not want to do such an intensive study. We did observed abnormal phase-shifting in the mice, which also is likely to contribute to the phenotype.

[Editors' note: further revisions were requested prior to acceptance, as described below.]

*Reviewer #1:*

*Although parts of the manuscript are much improved over the previous version, some issues would still benefit from further clarification as follows:*

*1) Second paragraph of Introduction section: "Mutations in mouse Fbxl3 or knockout of the Fbxl3 gene dramatically lengthens the period of mouse behavioral rhythms in constant darkness (Godinho et al., 2007; Hirano et al., 2013; Shi et al., 2013; Siepka et al., 2007), indicating that the protein stability of CRY2 is a critical determinant of circadian period in mice."*

*This is a misleading statement – all of the references cited studied the combined effects on CRY1 and CRY2 of Fbxl3. The authors should therefore say: "the protein stability of CRY1 and CRY2 is a critical determinant…".*

We have modified the description as suggested.

*2) Results section: "No other novel mutations were found in ~25 candidate circadian genes that were sequenced."*

*This statement is unclear. The Cry2 mutation studied is present in public databases (rs201220841), albeit at low frequency. Are there coding variations in the other sequenced circadian genes? If so, were these discounted based on their presence at a higher frequency in public databases, or by other criteria?*

This variant (rs201220841) was registered in May 2012 and March 2015 by two sequencing studies. Since we found this mutation before the registration of the database (2008) in a strong physiological candidate, we categorized the mutation as a novel variant and proceeded to generate the mutant mouse line. We looked through several hundreds of normal control subjects plus public genome sequences to confirm novelty in 2008. The recapitulation of a circadian phenotype led to us to the more detailed molecular analysis showing functional consequences in vitro. Since the reported variant in public databases is rare (0.008% in the ExAc database), and FASP is not uncommon (0.5% in general population, our unpublished data), it is not surprising that this variant exists at a low frequency in the general population (~120,000 in the ExAc database). This variant, therefore, explains a small but measurable proportion of FASP in the population. We have discussed this in the modified manuscript.

Regarding other candidate circadian genes, we did not find any other rare polymorphism co-segregating with the phenotype. Ultimately, the behavioral phenotype in the mouse model is what we rely on to confirm the causal relationship of the variant and behavioral trait.

*3) [Supplementary-material SD1-data] and Figure 1:*

*Although the subject IDs are not given in the pedigree, it seems based on the ages that subject 101374 would have to be the pro-band's nephew. In that case, he appears to be a carrier of the allele, although he has very late sleep times and a low H-O score. So, does the allele fail to segregate with the FASP phenotype in this case? At 21 years old, it seems doubtful that age is a sufficient explanation for this discrepancy. Penetrance and age of onset should be discussed.*

As pointed out by this reviewer, this subject’s phenotype cannot be categorized as FASP. However, in adolescence and young adult life, there is a well recognized tendency to be more phase delayed. When phenotyped at age 21, subject 101374 was at the statistical peak age of maximum phase delay and was prone to be even more phase-delayed by his male sex (Roenneberg T et al. A marker for the end of adolescence. Curr Biol 2004;14(24):R1038-9.).

In the recent authoritative review of advanced sleep phase in the International Classification of Sleep Disorders (American Academy of Sleep Medicine. International classification of sleep disorders, 3^rd^ ed. Darien, IL: American Academy of Sleep Medicine, 2014) there was insufficient peer-reviewed data on age-of-onset in Advanced Sleep Phase to even be mentioned, other than the trend toward higher prevalence in “older age.”

Thus, we have always considered individuals under age 30 who do not meet our strict diagnostic criteria to be “unknown”. It is likely that he will become more phase advanced as he approaches age 30 (still young enough so as not to be confused with the ASPS of aging). Because non-penetrance does occur, it is also possible that this individual will never manifest FASP. We have modified the discussion to reflect this important issue.

*4) Figure 2—figure supplement 1:*

*The authors describe a 'larger amplitude advance' on a Cry2-null background, but this is not clear from the figure. Have the results in the histograms been tested for statistical significance (as in Figure 2)?*

We did not detect significant differences in activity onset time and offset time due to the large variations, while there is a trend of advanced activity phase in hCRY2-A260T/Cry2 KO. However, we observed a significantly different activity and resting pattern (Figure 1, left) around ZT12-13 and ZT 22-0 to support the advanced sleep/wake phase. We didn’t have enough mice to do similar statistical analysis and agree that it is not clear how early the activity onset (immobility offset) is advanced in hCRY2-A260T/Cry2 KO mice from the profile data. We have modified the description in the main text accordingly.

*5) Figure 2; Figure 2—figure supplement 2:*

*Results on WT background are not quantified, so, the authors' claims on activity onset and offset times are hard to follow based on actograms from a single animal. In the supplementary figure, there appears to be a difference in activity offset in hCRY2-A260T-Tg compared to the hCry2-WT-Tg, but not the wildtype control?*

The activity offset was quantified and shown in Figure 2—figure supplement 2, middle panel. The activity onset was also quantified, but there was no significant phenotype (data not shown). We also show the activity profile in Figure 2—figure supplement 2 (left panel, average of 19 mice for each Tg line, and 7 mice for WT) as it is hard to interpret from actogram of a single animal.

As pointed out by this reviewer, mice carrying the hCry2-WT transgene showed delayed activity offset vs. WT littermates. We also found that a hCRY2 transgene (both WT and A260T) lengthened the active period (α) in DD (data not shown) and increased total wake time in LD (EEG data, data not shown), suggesting that hCRY2 overexpression affects sleep homeostasis. This leads to delayed activity offset in hCry2-WT-Tg mice vs. WT littermates. Thus, it is ideal that hCry2-A260T-Tg should be compared with hCry2-WT-Tg mice. See also comment 7 below.

*6) Figure 2—figure supplement 4:*

*Did the authors confirm whether the transgenes in fact rescues the Cry2 phenotype in the KO strain? The WT-Tg still seems to have a long period, is this actually different from the Cry2 KO background strain?*

In Figure 3, the circadian period of Cry2 KO (24.28 +/- 0.28 hr) was shortened by hCRY2-WT transgene (#1 24.1 +/- 0.00 hr; #2 23.65 +/- 0.18 hr). The circadian period of wheel-running activity of Cry2 KO (24.41 +/- 0.06 hr) was longer than that of hCRY2-WT/Cry2 KO mice (24.34 hr +/- 0.14 hr). We predicted that the expression of the human CRY2 transgene is much lower than that of mouse endogenous CRY2 and that the human wild-type CRY2 transgene partially rescued the Cry2 KO phenotype.

*7) Results section, “Shortened period and reduced phase-shift in hCRY2-A260T mice”: "Of note, there is no significant difference in the periods and phase-shifting of hCRY2-WT transgenic vs. transgene negative mice (Figure 2), indicating that the shortening of circadian period and abnormal phase-delay are not simply due to overexpression of hCRY2."*

*This is not the case for the activity offset in Figure 2—figure supplement 2, though?*

As pointed out by this reviewer, mice carrying the hCry2-WT transgene showed delayed activity offset vs. WT littermate. As described above (comment 5), we also found that a hCRY2 transgene (both WT and A260T) lengthened the active period (α) in DD (data not shown) and increased total wake time in LD (EEG data, data not shown), suggesting that hCRY2 overexpression affects sleep homeostasis. This leads to delayed activity offset in hCry2-WT-Tg mice vs. WT littermates. In the manuscript, we intended to mention that the circadian behavior (not sleep homeostasis) was not affected by the transgene.

*8) Figure 4—figure supplement 1: The quantification does not seem to match the shown WB at all. Rather, in the WB, the WT and mutant results seem essentially indistinguishable.*

We have repeated this experiment and inserted better quality figures more clearly demonstrating this point.